# scTab: Scaling cross-tissue single-cell annotation models

Felix Fischer[1,2], David S. Fischer[1,3], Roman Mukhin [4], Andrey Isaev [4], Evan Biederstedt[5,6,7,8], Alexandra-Chloé Villani[6,7,8,9] & Fabian J. Theis [1,2,10] ✉

Identifying cellular identities is a key use case in single-cell transcriptomics. While machine learning has been leveraged to automate cell annotation predictions for some time, there has been little progress in scaling neural networks to large data sets and in constructing models that generalize well across diverse tissues. Here, we propose scTab, an automated cell type prediction model specific to tabular data, and train it using a novel data augmentation scheme across a large corpus of single-cell RNA-seq observations (22.2 million cells). In this context, we show that cross-tissue annotation requires nonlinear models and that the performance of scTab scales both in terms of training dataset size and model size. Additionally, we show that the proposed data augmentation schema improves model generalization. In summary, we introduce a de novo cell type prediction model for single-cell RNA-seq data that can be trained across a large-scale collection of curated datasets and demonstrate the benefits of using deep learning methods in this paradigm.

Cell type annotation is a core step in the analysis of single-cell RNA-seq (scRNA-seq) data. Researchers typically examine prominent gene expression markers denoting a cell's identity and function, and assign a label based on a nomenclature that summarizes previously described cell types and states. While this task has been addressed in numerous analyses[1–4] and automated cell type prediction models[5–8], rigorously annotating new datasets remains a manual and time-consuming process. Moreover, given the confounding presence of technical batch effects and the inherent differences in quality across cells within datasets, the process of generating cell annotations remains unstandardized. These problems became especially pronounced in building comprehensive cell atlases in the Human Cell Atlas (HCA)[9], wherein unannotated datasets remain a bottleneck. Indeed, recent atlas-building efforts acutely highlight the challenges posed by both the lack of consensus in cell type annotations across datasets and how time-intensive it remains to standardize them[10,11]. A general model for

cell type annotation predictions − that is, a model trained on a large and diverse data corpus consisting of all human tissues in diverse − would assist with the atlas-building efforts of the HCA in several crucial ways: To begin with, such a model would lower the barrier of manually annotating datasets for researchers, offering suggestions with a standardized set of nomenclature. Predictions with a uniform set of vocabulary will naturally push the community to adopt consistent terms when referring to cell types. Moreover, such model suggestions will serve as hints, a baseline for researchers to modify and refine based on their own knowledge and expertise. Finally, such a process would allow scientists to annotate datasets at the scale required by the HCA and related initiatives.

Providing cell labels for unannotated datasets can be posed as a machine-learning classification task. However, cross-tissue classifiers trained on large-scale data collections that annotate cells from heterogeneous sources, irrespective of tissue of origin and assay type, are

[1]Department of Computational Health, Institute of Computational Biology, Helmholtz, Munich, Germany. [2]School of Computing, Information and Technology, Technical University of Munich, Munich, Germany. [3]Eric and Wendy Schmidt Center, Broad Institute of MIT and Harvard, Cambridge, MA 02142, USA. [4]eBook Applications LLC, Boston, MA 02467, USA. [5]Department of Biomedical Informatics, Harvard Medical School, Boston, MA 02115, USA. [6]Broad Institute of MIT and Harvard, Cambridge, MA 02142, USA. [7]Center for Immunology and Inflammatory Diseases, Massachusetts General Hospital, Charlestown, MA 02129, USA. [8]Krantz Family Center for Cancer Research, Massachusetts General Hospital, Boston, MA 02114, USA. [9]Department of Medicine, Harvard Medical School, Boston, MA 02115, USA. [10]TUM School of Life Sciences Weihenstephan, Technical University of Munich, Munich, Germany. ✉e-mail: fabian.theis@helmholtz-munich.de

slow to emerge. In this context, by cross-tissue annotation task we mean training and evaluating a single classifier model on a diverse selection of tissues. This is in contrast to organ-specific classifiers which are trained and evaluated only on a single tissue or only a narrow selection of closely related tissues. Current models often address only specific scenarios and do not focus on strong generalization capabilities beyond the datasets they are trained on[12], partly due to fragmented data collection efforts. Recent large data curation efforts streamline the training of generalist models because they expose cell type labels and other metadata in structured vocabularies of ontologies and consistent feature spaces[13,14]. In particular, CELLxGENE hosts a draft of a curated data collection that allows for models to be trained across a significantly larger range of datasets than what was possible before[15]. Nevertheless, a dominant paradigm of the cell type annotation task has been annotation transfer[16], specifically the approach of projecting entire samples of cells on an annotated reference atlas to transfer cell type labels[1,17,18]. Annotation transfer is often used in organ- or lineage-specific scenarios, yet it is critically limited by the quality and similarity of the annotated reference. In contrast to this paradigm of query-to-reference mapping, we focus on the problem of general cell type annotation which entails training models across a large-scale data corpus to predict cell type labels solely based on gene expression. Here, we would like to add a clarification note regarding the distinction between cross-tissue and general cell type classification: with both terms, we essentially refer to the same task; namely, directly predicting cell type labels solely based on the gene expression of a cell with a single unified model that works across a wide range of tissues. However, this task is not fully solved yet as the data and the quality of cell type annotations are just not there yet. To highlight this we used two different terms: Cross-tissue annotation refers to the general task; General cell type annotation refers to the overall goal we want to achieve - one might call it the holy grail of cell type annotation. It's also worth noting that the predictions of such a general cell type annotation model would be community-driven; researchers would not be required to rely upon annotation transfer-based methods. Such approaches inevitably suffer from model overtraining upon a single reference (often with lab-specific cell labels) and strongly encourage researchers to choose a context-specific reference close enough to their study of interest as a basis for trustworthy cell annotations.

Several aspects of the general cell type annotation problem remain ambiguous: Firstly, initial attempts to increase model complexity in cell type annotation to improve classification performance have failed to improve over linear baseline models[7,19,20]. Consequently, the question pertains to whether large-scale, non-linear models learn cell state representations that are more useful for this classification task than linear, well-tuned baseline models that are trained on large-scale datasets as well. Furthermore, recent efforts have started to use large-scale data corpora with tens of millions of scRNA-seq profiles to train deep learning models[21–23]. However, those efforts on foundation models mainly focus on learning cell representations or embeddings in an unsupervised manner, without a specific focus on cell type annotation (and especially without making use of author-provided cell type labels). Indeed, the cell annotation tasks considered in these efforts are either only fine-tuned to specific cell type classification problems with only a few cell types[21,23] or do not directly predict cell type labels but rely on finding similar cells in an annotated reference[22]. Moreover, these initial attempts at building foundation models that cover a large data corpus largely treat the cell type labels as mutually exclusive and ignore label relations that were previously exploited in organ-centric classification tasks[14,24], thus questioning if the resulting benchmarking metrics are faithful evaluations of the performance on these heterogeneous datasets. Furthermore, according to recent benchmarks[25,26], recent foundation models often only show comparable performance to simpler and often linear reference models in the zero or few-shot setting. Zero-shot setting means training a linear classifier based on the

embedding obtained from a pre-trained foundation model, and few-shot setting means fine-tuning a pre-trained foundation model to a specific task based on a small example dataset. Lastly, it remains unclear how cross-tissue cell type classifiers compare their respective organ-specific counterparts. Here, we address these challenges by assembling a benchmark dataset for cross-tissue cell type classification and carefully analyzing a cross-tissue cell type classifier optimized for cell type annotation on tabular scRNA-seq data: scTab. scTab uses observation-wise feature attention to reduce the number of input features for each observation. This helps the model to be more robust to overfitting to poorly generalizable features in the training data, which is often an issue for tabular data as there is no prior knowledge about the underlying structure of the data - unlike e.g. for images or text[27].

Modern deep learning builds upon the idea of learning a decision function purely based on data. This idea brought stunning breakthroughs in the fields of computer vision and natural language processing, where the approach of using big models and large-scale training datasets to regularize those models drastically outperforms other traditional (usually feature-engineering-based) machine learning approaches. Hence, we leverage well-defined benchmark metrics[5,6] for cell type classification to understand the performance of deep learning models trained on large scRNA-seq data corpora, focusing on the scaling behavior of such models with respect to the training data size as well as the model size[28]. We find that analogous to the work in computer vision, cell type classification from scRNA-seq data substantially benefits from large-scale training of deep-learning-based models[29], and model generalizability can be improved by artificially increasing the training data size through data augmentation[30]. In addition, we find that by scaling cell type classification to large-scale datasets, deep-learning models outperform their linear counterparts, in contrast to what was reported before[7]. Yet, well-defined baseline models are still relatively powerful, thus suggesting caution in the design of benchmarking experiments in foundation models. In summary, our detailed analysis demonstrates the strengths of deep learning-based approaches over their linear counterparts in large-scale, cross-tissue cell type classification and shows that classification performance scales both with respect to training data and model size.

## Results

### A dataset and evaluation metric to study the scaling behavior of cross-tissue cell type classification models

We set out to build a dataset on which a cross-tissue cell type classification model could be trained and evaluated. In the existing literature, we identified three approaches to creating such a dataset. The first approach involves assembling study- or organ-specific datasets and homogenizing cell type labels to obtain a mutually exclusive set of labels or a tree with levels of mutually exclusive labels[5–7]. The second approach centers around assembling organ-specific datasets and mitigating annotation granularity differences by using the Cell Ontology[31,32] to establish a directed acyclic graph between the observed labels[14]. The third approach entails the collection of an organism-wide data corpus with ontology-constrained labels treated as mutually exclusive, thus ignoring the hierarchical structure of labels in the ontology[21,23]. This is problematic since the hierarchical dependency between labels is a key structure of these datasets, e.g. a CD4-positive, alpha-beta T cell is also a T cell and a lymphocyte, and penalizing a model that predicts CD4-positive, alpha-beta T cell for a cell that is labeled as a T cell results in an evaluation that is biased towards the model being able to mimic the annotation granularity of the data instead of its ability to distinguish cell types. Here, we leveraged the cell type relations given by the Cell Ontology[31] across a data corpus of all human tissues, using a release of the cell census by CELLxGENE as a root dataset (Methods). This data corpus reflects a large number (164) of cell types in the human body, therefore, we refer

to the resulting label prediction problem as general cell type classification.

To benchmark a cell type prediction task that represents the recent literature closely[6,19], we considered models with a softmax-constrained output across all observed cell type labels. To account for differences in annotation granularity across datasets, we adjusted predictions based on the Cell Ontology to not incur penalties if a model predicts a more fine-grained label than the original author annotation (Methods). We modified the original release of the cell census in preparation for this task (Methods). First, it is important to realize that public data corpora are not necessarily deduplicated. Often, cells are present in a primary dataset that originates from an original study and are also contained in secondary datasets (metastudies), such as atlas datasets. We only considered instances of cells in primary datasets to avoid data leakage through duplicate cells. Second, we removed cells with broad cell type labels, using a heuristic that removes each cell type with less than seven parent nodes in the Cell Ontology. (Note: we would like to emphasize that the previously described filtering step is merely a heuristic, i.e. there are still coarse cell type labels present in the scTab data corpus even after our filtering heuristic, given the respective detail or sparsity of terms within the Cell Ontology for certain tissues.). For the remaining cell types, we used the most fine-grained labels given by the author associated with the CL term as specified by the CELLxGENE schema and did not map the author-provided cell types to more coarse Cell Ontology terms. Third, we restricted to cells measured with the most common group of 10X Genomics technology-related assays, to reduce the strength of confounding sources of variation in the dataset. Fourth, we removed cells from rare cell types with less than 5000 instances or those present in less than 30 donors to accurately assess how the trained classifiers generalize to unseen donors (Methods) (Note: these thresholds are dataset dependent and should be adjusted accordingly for smaller datasets). The resulting dataset contains 22.2 million cells, with 5052 donors and 164 cell type labels (Fig. 1a). We defined test holdouts based on donor annotation, which we see as a sensible compromise between an entirely random split and a split based on holdout studies. The donor-wise split improves the coverage of labels in both training and test sets compared to a split based on studies, and reduces leakage of similar observations between training and test data compared to a random split of cells, thus creating an evaluation set that is better suitable to assess the generalization capabilities of a classifier. As a benchmarking metric, we chose the macro-averaged F1-score (macro F1-score) (Methods) to account for class imbalances and to give each cell type an equal weight in the overall score.

## A feature-attention-based, scalable, deep-learning model for cross-tissue cell type classification

Studying the scaling behavior of deep-learning-based models necessitates a scalable model implementation that can be trained on bigger-than-memory datasets. In addition, we ask if recent extensions beyond classical multi-layer perceptrons (MLP) improve prediction as they have in other fields. Since gene expression profiles are not ordered, we decided against sequence-based models such as transformers[21,23] and instead selected a recent architecture specifically proposed for tabular data[33]. Here, we introduce scTab (Fig. 1b), which is a scalable implementation of the TabNet architecture[33], which we adapted to the single-cell use case: scTab is specifically designed for the tabular structure of scRNA-seq data through the use of feature attention, which enables the network to focus its model capacity on more reliable input features. After normalization, it encodes data via a feature transformer and selects relevant input features through feature attention via an attention transformer block (Methods). We modified the original TabNet implementation in a few crucial ways: scTab's input data assumption is adapted to the single-cell setting, in particular, the input gene expression is size factor normalized to 10,000 counts per

cell and log1p transformed. This common normalization for scRNA-seq data[6,22] cannot be replicated by the simple batch normalization layer used in the original TabNet architecture. We additionally modified the original TabNet architecture to improve computational efficiency, namely by reducing the number of feature and attention blocks (which we found unnecessary after profiling), and training dynamics for faster convergence (Methods). For better model generalizability, we further added a data augmentation step as described later below. Finally, scTab quantifies prediction uncertainty using empiric uncertainty probabilities based on deep ensembles[34] (Methods).

## Cross-tissue cell type classification requires nonlinear models

To showcase the performance of state-of-the-art models according to recent benchmarks[5,19], we first retrained a CellTypist model[6] (Methods) to a random subsample of our training corpus. The current CellTypist implementation necessitated the full training data to be subsampled for model training as on the one hand it requires all the training data to be loaded into memory and on the other hand it lacks GPU acceleration. Here, we subsampled to 1.5 million cells. This re-trained reference model achieved a macro F1-score of $0.7304 \pm 0.0015$ ($\pm$ is indicating the standard deviation across several model fits with a different random initialization each) (Fig. 1c). Given this performance of the reference CellTypist model, we investigated if performance could be increased by scaling logistic regression-based models to take advantage of the full training data size. We implemented a logistic regression-based model not subject to these limitations (Methods) and trained this model with a cross-entropy loss. This model outperformed the CellTypist reference model and achieved a macro F1-score of $0.7848 \pm 0.0001$ (Fig. 1c), showing the potential of scaling existing linear models to take advantage of larger datasets. Having optimized the linear reference model, we benchmarked three nonlinear models against this baseline: our scTab model, an MLP previously proposed for this task[7,14] - but found to not outperform linear models within single tissues - and an XGBoost model that reported robust performances on classification tasks for tabular data[27]: The nonlinear models outperformed the linear model ($0.8295 \pm 0.0007$ macro F1-score for scTab (fitted with data augmentation), $0.8127 \pm 0.0005$ for XGBoost, $0.7971 \pm 0.0012$ for MLP (fitted with data augmentation)) (Fig. 1c, Supplementary Table 1), demonstrating that cross-tissue cell type classification is complex enough to benefit from nonlinear models. Moreover, scTab outperformed the linear model on all organ systems when these were considered separately (Fig. 1d). Similarly, we compared scTab with the single-cell transformer model scGPT without fine-tuning (zero-shot setting)[23] i.e. training a logistic regression model on the scGPT embeddings (Methods, Supp. Fig. 1), which achieved a significantly lower macro F1-score of $0.7301 \pm 0.0035$ (Supplementary Table 1), comparable to PCA embeddings. Moreover, we fine-tuned a scGPT model to our training data; the fine-tuned model achieved a macro F1-score of 0.749 (Supp. Fig. 1, Supplementary Table 1). Lastly, we benchmarked against CIForm[35] - another transformer-based model for cell type annotation. The CIForm model achieved a macro F1-score of 0.766 (due to the large memory requirements of CIForm we had to subsample the training data to 750,000 cells). Lastly, we show a comparison of the training and inference times in Supplementary Table 2.

Now, looking in more detail where the performance improvement of scTab stems from, one can observe that scTab performs particularly well in distinguishing between closely related classes of refined cell subtypes. This is most prominent when looking at T cell subtypes (Supplementary Fig. 2) for which the Cell Ontology is especially detailed, thus making it possible to resolve such fine-grained differences between subtypes. Notably, this trend is in fact not only limited to T cells but can also be observed in other cell lineages of the immune system, namely, B cells, monocytes, macrophages, and granulocytes. (Indeed, for many immune cell lineages, the Cell Ontolgy is quite

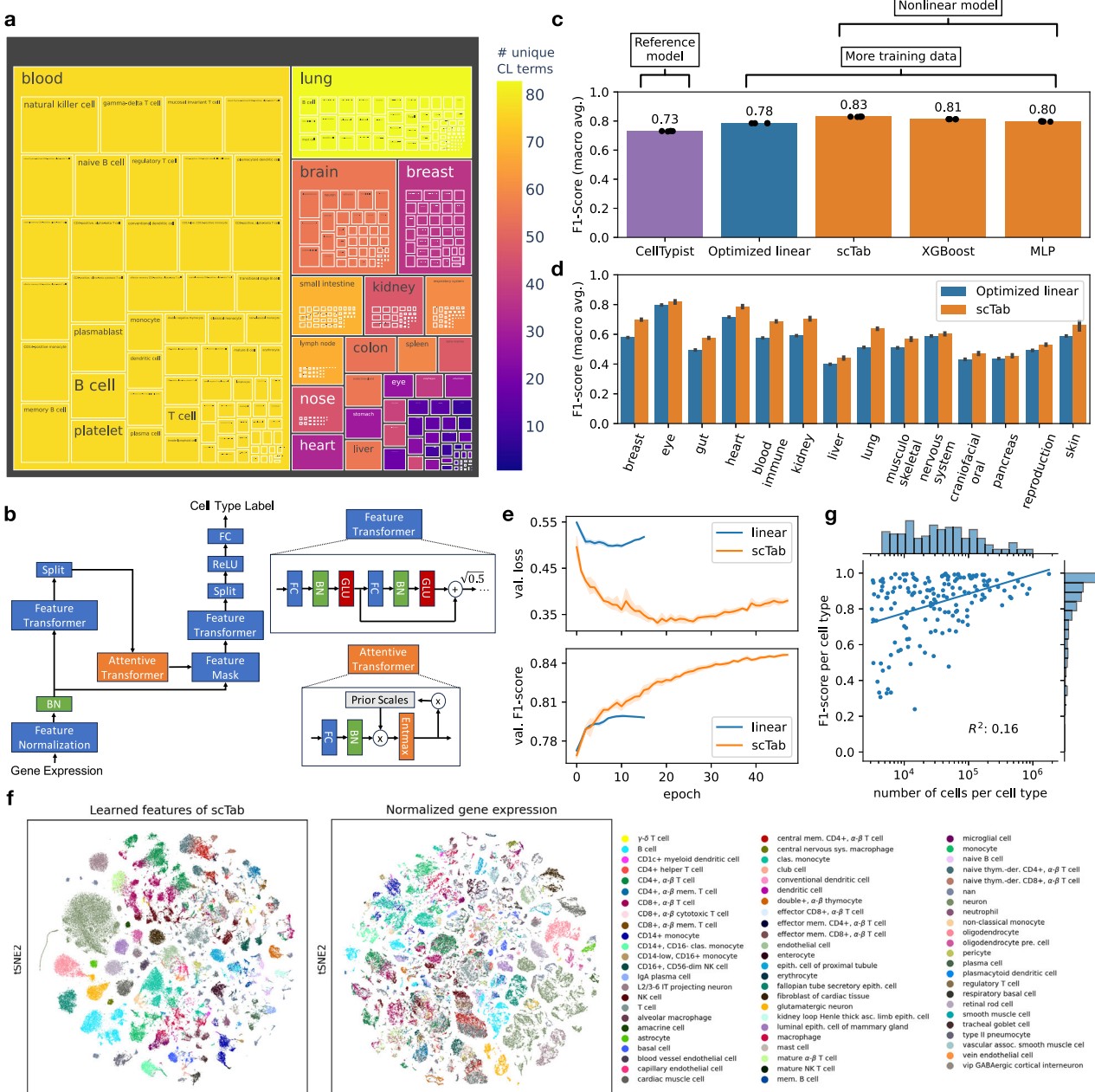

**Fig. 1 | scTab enables organism-wide, scalable, and robust cell type classification on single-cell RNA-seq data. a** Treemap plot showing the dataset composition across cell types and tissues. The outer rectangles correspond to the number of donors per tissue, the inner boxes correspond to the number of donors for each cell type and the color scale highlights the number of unique cell types per tissue (Supp. Fig. 5b shows the number of unique cell types grouped by Human Cell Atlas bionetworks). The dataset spans 22.2 million cells, 5,052 donors, 249 datasets, 164 cell types, and 56 different tissues. See Supp. Fig. 5 for more detailed summary statistics of the scTab data corpus. **b** scTab architecture (Methods): after input feature normalization, scTab encodes data via a feature transformer and selects relevant input features through feature attention. (FC: fully connected layer, BN: batch-norm layer, GLU: gated-linear-unit, ReLU: rectified-linear-unit). **c** Comparison of classification performance (macro F1-score) of linear reference models (CellTypist, subsampled to 1.5 million cells), optimized linear) and nonlinear models (scTab, XGBoost, MLP (multi-layer perceptron)). Data are presented as mean values ± SD. Source data are provided as a Source Data file. **d** Classification performance (macro F1-score) grouped by organ system of scTab and the optimized linear model. Data are presented as mean values ± SD. Source data are provided as a Source Data file. **e** Cross-entropy loss and macro F1-score on the validation set plotted after each epoch for scTab and the optimized linear model. Data are presented as mean values ± 95% CI. **f** tSNE plots of raw features and the learned features of scTab with the top 70 most frequent cell types superimposed on the holdout test set. **g** F1-score per cell type plotted against the number of unique cells observed per cell type for scTab. The histogram on the y-axis shows the distribution of F1-scores and the histogram on the x-axis shows the distribution of unique cells per cell type.

detailed, and therefore these subtypes existed in the training data.) To assess the above trend quantitatively, for both scTab and the optimized linear model we compared the classification performance of identifying cells associated with a specific cell lineage against the classification performance of distinguishing fine-grained subtypes

associated with the respective cell lineage. Comparing the difference in classification performance of the two models for those two settings, one can see that both models achieve similar performance when identifying cells of a particular cell lineage but that the performance difference becomes more pronounced when distinguishing fine-

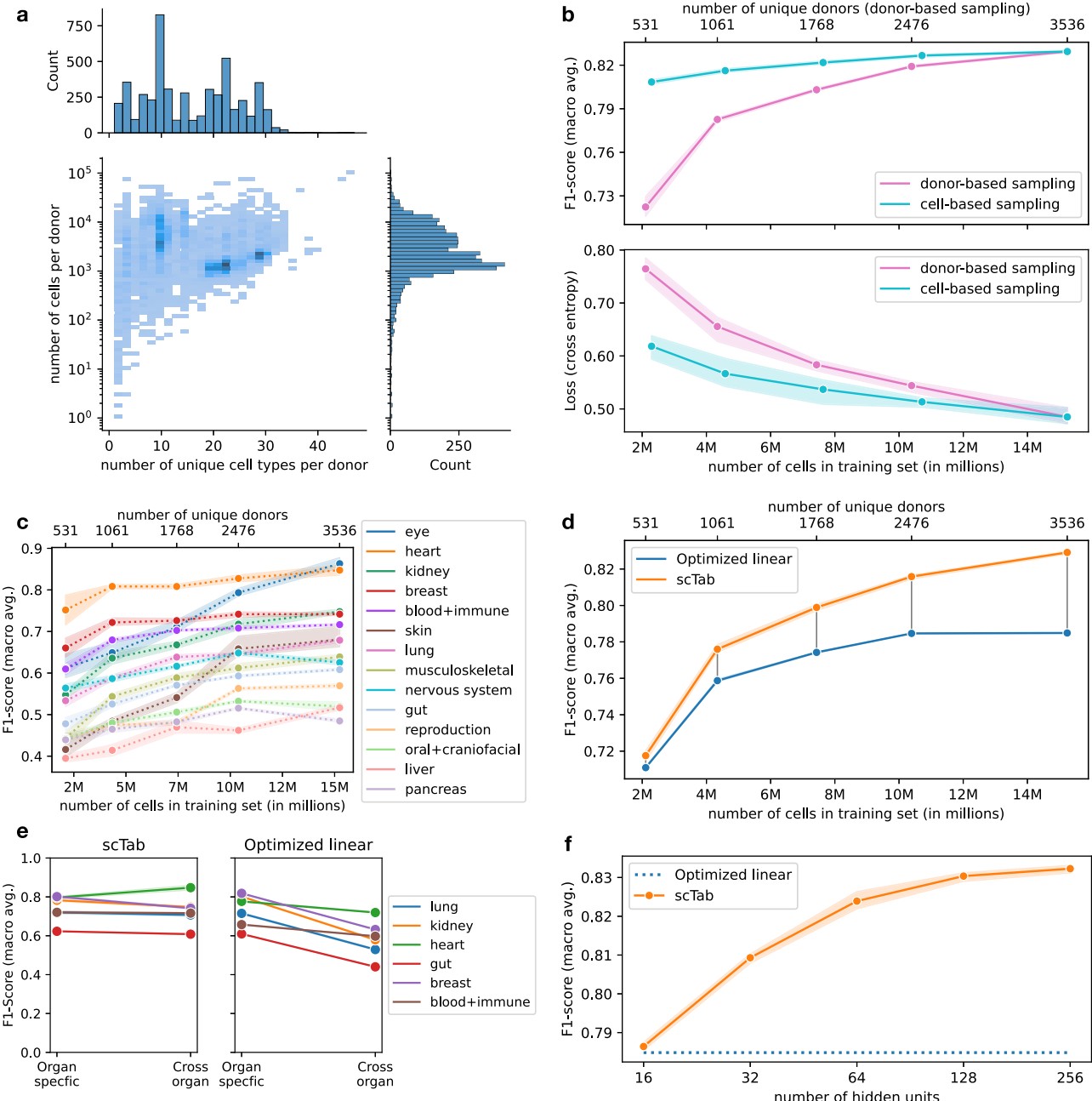

**Fig. 2 | Non-trivial scaling behavior of scTab in cross-tissue cell type prediction.** **a** Distribution of donors with respect to the number of unique cell types (x-axis) and with respect to the number of cells (y-axis). The y-axis histogram shows the distribution of donors with respect to the number of cells (log scale). The x-axis histogram indicates the distribution of donors with respect to the number of unique cell types. **b** Scaling behavior of scTab with respect to the size of the training data for two simulated scenarios in terms of macro F1-score and cross-entropy loss: i) cell-based subsampling which corresponds to increasing the number of sequenced cells while keeping the observed biological diversity constant ii) donor-based subsampling which corresponds to increasing the observed biological diversity. All cell types from the test set were observed during model training for all subsampled datasets. Data are presented as mean values ± 95% CI. Source data are provided as a Source Data file. **c** Scaling of the cross-organ model from Fig. 2b with

respect to training data size grouped by organ system (subsampling is done based on donor-based subsampling). Data are presented as mean values ± 95% CI. Source data are provided as a Source Data file. **d** Scaling behavior of scTab versus our linear reference model with respect to the training data size. Data are presented as mean values ± 95% CI. Source data are provided as a Source Data file. **e** Effect of training only on organ-specific data versus training on cross-organ data on organ-specific classification performance (evaluated on test data subset only to the corresponding organ) for scTab and the optimized linear model. Data are presented as mean values ± 95% CI. Source data are provided as a Source Data file. **f** Scaling behavior with respect to model size. The number of hidden units refers to the size of the fully connected layers (FC) in the architecture (Fig. 1b, Methods). Data are presented as mean values ± 95% CI. Source data are provided as a Source Data file.

grained subtypes for scTab (Supplementary Fig. 3a). This example highlights the potential of our scTab model to distinguish between closely related cell types and hints that the improvement of non-linear models such as scTab will further increase once more finely annotated training data becomes available, and the Cell Ontology incorporates

more finely-grained subtypes of cells in other tissues. It is perhaps unsurprising how relatively detailed immune cells are in the Cell Ontology with respect to other systems. Moreover, this trend does not seem to be limited to only the immune system but can be observed in other cell lineages outside of the immune system as well

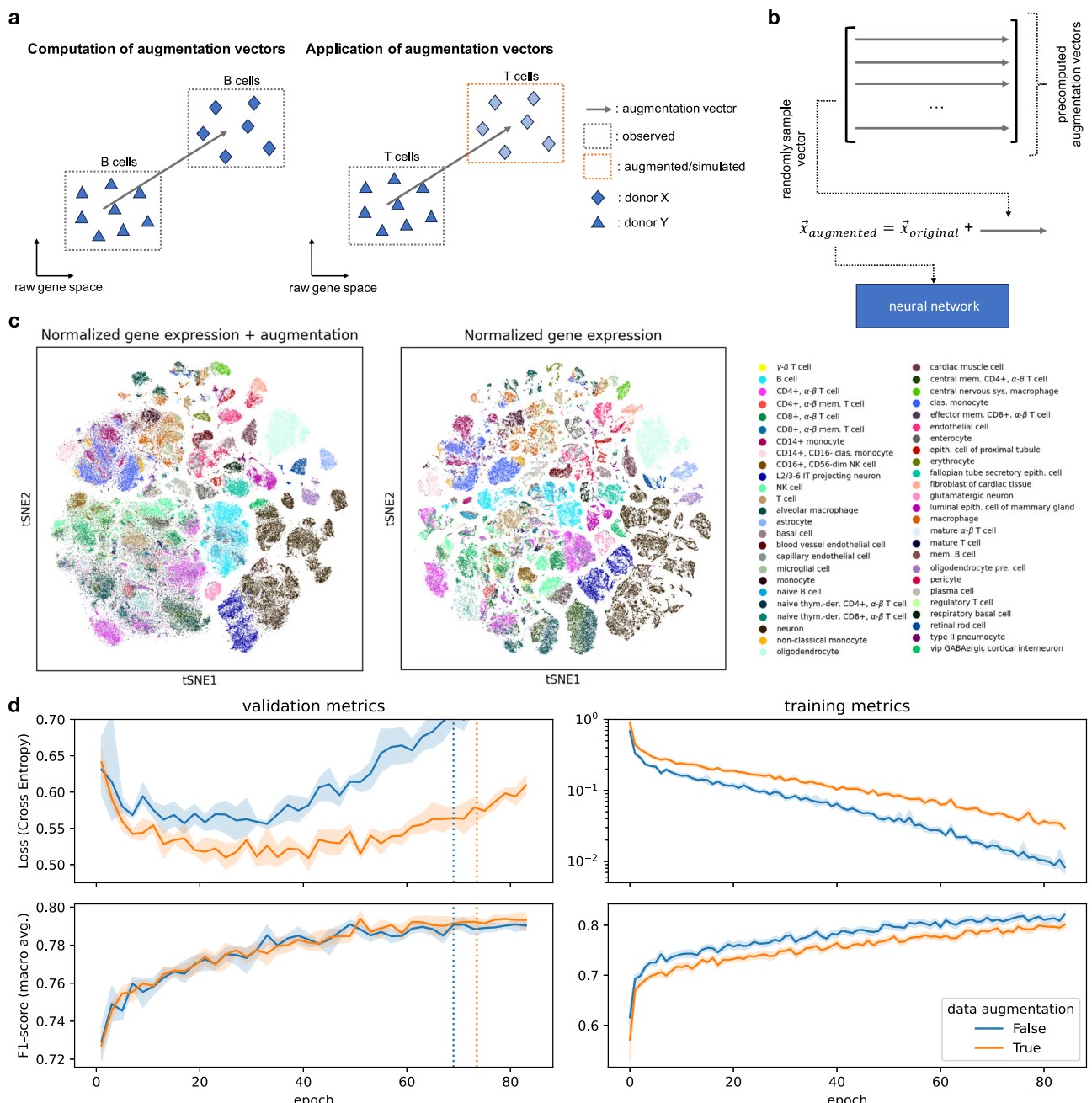

**Fig. 3 | Data augmentation for scRNA-seq cell type classification improves model generalizability. a** Illustration of the data augmentation procedure. The difference vector in raw gene space between the same cell type observed across two donors can be used to simulate how the gene expression of a cell type might look for a different donor and, thus, artificially increase the training data size. **b** For each input vector to the neural network, an augmentation vector is randomly sampled and added to the original input vector. The augmented vector is then fed into the neural network (due to simplicity the batch dimension is omitted in the sketch). **c** tSNE visualization of original and augmented data. One can see that the augmentation blurs out the boundaries of the cell types but that the main source of variation (cell type) is still preserved. **d** Effect of augmentation on training and validation loss and macro F1-score (training data was subset to 4.3 million cells (Methods)). One can observe the desired effect of data augmentation, an increase in training loss (regularizing effect), and a decrease in validation loss. The dashed vertical lines indicate how long the models with and without data augmentation are fitted on average (early stopping is done based on the macro F1-score), respectively. Data are presented as mean values ± 95% CI.

(Supplementary Fig. 3b). Nonetheless, we would like to emphasize here that this trend does not hold in general for all tissues. Cell types related to the brain are the most notable exception here, which we hypothesize is probably influenced by the relative lack of resolution of the Cell Ontology for brain-related cell types. Related to the previous point, we asked how our scTab model would perform on more coarse cell type labels (Supp. Fig. 7a). To do so, we mapped the 164 original cell type labels to 31 coarse cell type labels (Methods). Evaluated on the

31 coarse cell type labels, scTab achieves a macro F1-Score of 0.897 compared to 0.830 on the fine-grained annotations. This trend is not surprising, as we expect classification between coarse categories of cell types (with distinct transcriptional signatures) should be easier than classification between refined subtypes of cell types (which share similar molecular signatures).

Further to this, when looking at uncertainty scores calculated based on deep ensembles[34] (Methods), one can see that incorrect

predictions on the holdout test data and novel cell types, which are not present in the training data, are also associated with a higher model uncertainty (Methods, Supp. Fig. 4). Besides the performance advantage over linear models, scTab also showcases different training dynamics: On the one hand, the difference to the linear model is more pronounced when looking at the loss, and on the other hand, scTab is trained for more epochs (Fig. 1e). When qualitatively inspecting the representations learned by scTab in a tSNE plot, we found that cell types show consistent separation and that the latent space was more structured compared to the raw feature space (Fig. 1f). Moreover, similar cell types group together in the latent space (Supp. Fig. 5a).

Another important observation is that classes on which errors were made tended to be those that were represented by few observations (Fig. 1g). This further motivates that classification performance can indeed be improved by adding more training examples specifically for cell types that are hard to classify. When looking at the plot in more detail, one can see that some cell types achieve quite remarkable classification performance despite very little available training data. To investigate this further, we looked at the cell types that have an F1-score of 0.9 or higher and that have less than 10,000 cells in our data corpus (Supplementary Table 3). Looking at the table, one can see that this phenomenon appears to be enriched among cell types associated with the central nervous system. However, it's not limited to those. Thus, we hypothesize that there is a more general explanation for this. Namely, those cell types have a specific gene signature that on the one hand makes the cell types easy to distinguish from other cell types and on the other hand generalizes well from the training to the test data. To investigate this hypothesis in more detail, we look at the feature attention masks of the scTab model to determine which features are important when generating predictions. The feature attention masks are sparse (they are non-zero for only around 1% of the gene space) and indicate which genes the model prioritizes when classifying a specific cell or cell type. We selected the top 200 genes (~1% of the total gene space) ranked by the feature attention scores of the scTab model and looked at how well a linear model can separate a cell type from all other cell types purely based on those top 200 genes. We then compared the separation scores (measured by the area under the precision-recall-curve) to a reference group that included all cell types with an F1-score of less than 0.4 and again less than 10,000 cells of available training data (Supp. Fig. 6). One can see that the easy-to-predict cell types are easier to distinguish based on the selected top 200 genes and that, more importantly, the gene signature generalizes well from the training to the test data (for the easy-to-predict cell types the F1-scores only drop around 0.03 versus 0.32 for the reference group). Moreover, we checked whether the genes in the feature attention masks do indeed correspond to the biology of the predicted cell type. We found that these gene signatures appear not to be random. The top 25 genes as ranked by feature attention scores were often found to be biologically relevant: as a brief example, the top genes for predicted brain cells include genes encoding neurexin and synaptotagmin proteins (e.g. *NRXN1*, *SYT1*), the top genes in the feature attention masks for all predicted macrophages included genes related to the immune system (e.g. *FCGR3A*, *HLA-DRA*), and so forth.

Lastly, it is interesting to investigate how scTab, which is only trained on 10X-based training data, would perform on data from non-10X-based sequencing protocols. Hence, we looked at the classification performance (measured by macro F1-score) grouped by sequencing protocol (Supp. Fig. 7b). One can see that scTab achieves decent classification performance on about half of the non-10X-based sequencing protocols with a macro F1-Score of ~0.4. In comparison, the macro F1-Score on the holdout 10X-data is ~0.8. This is quite impressive, given that the model has to generalize to unseen datasets that are measured with a different sequencing protocol - meaning there is a much stronger shift in data distribution compared to the donor-based holdout evaluation setting. However, there are also sequencing protocols to which our model generalizes quite poorly (namely STRT-seq, Smart-seq2, and BD Rhapsody Targeted mRNA), indicating an even stronger shift in data distribution for those sequencing protocols. The above observations further justify our decision to limit the training data only to 10X-based sequencing protocols. Given the strong shift in data distribution between the different sequencing protocols and that the data from CELLxGENE is clearly dominated by 10X-based sequencing protocols (Supp. Fig. 7c), it would make it challenging to train and evaluate our model on non-10X-based data reliably.

## Cross-tissue cell type classification scales with dataset and model size

A key driver behind the success of deep-learning-based models in computer vision or natural language processing is their ability to take advantage of larger datasets. This scaling behavior was a driving factor of recent advances in computer vision and natural language processing and led to the study of how model performances scale with model size and training examples[28,29]. Having established that the cross-tissue cell type classification problem satisfies this premise, we next set out to study its scaling behavior. In contrast to images, the heterogeneity of a scRNA-seq corpus is not trivially measured by the number of observations (cells). Some datasets contain densely sampled cell states, in which new samples would simply replicate what is already captured, whereas other samples are relatively sparsely sampled (Fig. 2a). To account for this complexity in the study of data scaling behavior, we compared the test performance of models trained on subsets of the full corpus, either subsampled randomly by cells as a control, or subsampled by donors to tie the subset size closer to the relative complexity captured by this dataset (Methods). Indeed, we observed a strong scaling of the loss and macro F1-score with respect to the dataset size for donor sub-sampling, but a much weaker scaling for cell-subsampling (Fig. 2b) indicating that scaling with respect to the training dataset size is mostly driven by batch diversity rather than the number of cells. This scaling also held for all organ systems when inspected individually, with a minimum difference in macro F1-score of $0.0454 \pm 0.0083$ between a dataset of 2.1 million cells and the full training dataset of 15.2 million cells, and a median difference of $0.1219 \pm 0.0212$ (Fig. 2c). A potential reason for the difference in classification performance between different organ systems lies in the number of observed cell types per organ system - the F1-scores have a correlation of $-0.55$ ($p$-value: 0.035) with the number of observed cell types per organ system. The observed scaling with data size was also specific to scTab and was not exhibited by the linear model (Fig. 2d), whose learning curve flattens out earlier, resulting in an improvement of the macro F1-score by scTab over our linear reference model of $0.0447 \pm 0.0008$ when using the full training data. Moreover, we compared the organ-specific cell type classification performance of models trained only on organ-specific data against their cross-organ counterparts. We did this evaluation for the organs that showed the biggest difference in classification performance between scTab and the optimized linear model (see Fig. 1d), namely: the lung, kidney, heart, gut, breast, and blood+immune. The deep-learning-based scTab model shows more robust performances compared to its linear counterpart (Fig. 2e) across all investigated organs, meaning the classification performance is a lot less affected when scaling to the cross-organ setting, suggesting that cross-organ cell type classification can benefit from using non-linear models. Naturally, we must emphasize that inconsistencies between author-provided cell annotations across tissue systems (and individual researchers) remain a major confounding factor for performance tests above. These are to be corrected as the community resolves existing disagreements behind how certain cell types are defined, along with the arrival of high-quality reference atlases.

Focussing on assessing model capacity at increased data size, we performed a second scaling experiment in which we kept the full dataset but compared scTab implementations with different numbers of parameters (Methods). We found a significant improvement in performance between the smallest model with 1.7 million parameters ($0.7864 \pm 0.0010$ macro F1-score) and the largest model with 16.2 million parameters ($0.8323 \pm 0.0010$ macro F1-score) (Fig. 2f). Overall, the above-mentioned differences in scaling behavior to a baseline linear model show that a nonlinear model is better able to take advantage of larger and more diverse data sets and that it is able to model complex non-linear relationships.

## Data augmentation improves classifier generalizability

Due to their high model capacity, deep learning models are known to easily overfit the training data and can even fit random labels[36]. One well-established technique to reduce overfitting and thus improve model generalizability is to artificially increase training data size by applying semantically preserving transformations. For images, these transformations include rotating or cropping input images during training[30]. Data augmentation serves as a regularization technique that yields models with better generalization capabilities and less impacted by dataset-shift phenomena. So far, data augmentation has not been consistently applied in single-cell genomics, due to the limited capacity of most scRNA-seq models, and due to the lack of sensible augmentation strategies. Here, we propose a augmentation strategy for scRNA-seq data and evaluate it for cell type prediction with our scTab model. Notably, our data augmentation strategy is not only limited to scTab but can be used in combination with other models as well. The motivation for the proposed data augmentation is to simulate the gene expression vector of a target cell if it were observed in a different donor. To do so, we precompute augmentation vectors based on the training data that can be added to the original gene expression vectors during model training (Fig. 3b). The data augmentation vectors are the average difference computed in the full gene space between cells of the same cell type observed in two different donors. Thus, by adding those augmentation vectors to the gene expression vector of the original cell, one can simulate the gene expression vector of a target cell in a different donor, extending the training data domain in these incompletely observed donors (Fig. 3a). Before evaluating the effect of this augmentation on model fits, we established that it did not severely disrupt the training data structure. Boundaries between cell types are blurred in a tSNE of the augmented data. Still, cell type identity as a main source of variation in the data is preserved (Fig. 3c) as quantified by a similar variance decomposition in terms of cell type and donor labels ($R^2 = 0.189$ for the raw data, $R^2 = 0.164$ for the augmented data, Supplementary Table 4, Methods). We found this augmentation strategy to regularize models, training loss increased upon using augmentation, and the macro F1-score on the training data decreased. Model generalization was improved on the validation set as measured by reduced loss and increased macro F1-score (Fig. 3d). When looking at the holdout test set, the proposed data augmentation significantly reduces the loss from $0.797 \pm 0.05$ to $0.659 \pm 0.04$ (p-value: 0.0039) and significantly increases the macro F1-score from $0.7755 \pm 0.0020$ to $0.7841 \pm 0.0030$ (p-value: 0.0016) (Supplementary Table 5). These results show that sensible data augmentation techniques for scRNA-seq data can significantly improve the generalization performance of cross-tissue cell type classifiers.

## Robust benchmarks for cross-tissue cell type classification

It is common practice in machine learning to have standardized large-scale benchmark data sets such as the ImageNet[37] subset for the ImageNet Large Scale Visual Recognition Challenge[38] and the Microsoft COCO dataset[39] in computer vision, the GLUE/SuperGLUE dataset[40,41] and the WMT2014 English-German datase[42] in natural language processing. These benchmark datasets enable models to be trained on bigger data corpora and allow for structured model benchmarks that usually do not require re-training of reference models. Such ready-to-use and large-scale benchmark datasets for cell type classification on single-cell transcriptomics data are not yet easily accessible. Creating such datasets for scRNA-seq data comes with two key challenges: On the one hand, such datasets need to come with a performant and easy-to-use data-loading infrastructure, that can scale to bigger-than-memory datasets. Otherwise, it becomes challenging for users without the proper technical background to use such datasets in their workflow. On the other hand, such datasets should be predefined, easily accessible, and come with fixed training, validation, and test splits to make results easily comparable. Now, to encourage similar practices, our processed benchmark dataset with predefined train, validation, and test splits and the accompanying data loading infrastructure is available to download (Methods). The downloadable dataset is ready to use out-of-the-box with an efficient data loader (Supp. Fig. 8, Methods). Furthermore, the dataset comes with a set of well-tuned reference models (Methods) that can be directly used for further benchmarking efforts. The need for well-tuned reference models is demonstrated by the comparison of the performance of the XGBoost and CellTypist models given default parameters and their respective performance given tuned parameters. On the benchmark data, the performance, measured by macro F1-score, could be increased from $0.5855 \pm 0.0112$ to $0.8127 \pm 0.0005$ for the XGBoost model and from $0.6258 \pm 0.0036$ to $0.7304 \pm 0.0015$ for the CellTypist model respectively (Supplementary Table 6). We expect this combination of a well-defined benchmark dataset with well-tuned baseline models to facilitate the systematic study of model scaling laws, which are of importance for the establishment and evaluation of foundation models[21–23,43,44].

## Discussion

We introduced cross-tissue cell type classification on a whole-body human data corpus of scRNA-seq data as a machine learning task that facilitates cell type annotation and that can benefit from large-scale data collections and the usage of larger, non-linear models similar to examples in computer vision[29]. Most notably, we showed the potential of our non-linear scTab model over linear models when distinguishing between fine-grained subtypes of cell types. Moreover, we demonstrated scaling of model performance with training dataset size and model size on this task, noting that batch diversity dominates the raw number of cells in this data scaling. We also found that model overfitting can be mitigated through data augmentation. Additionally, the analysis and models introduced here provide a reproducible context for future work on cross-tissue cell type classification which is a cornerstone in the context of foundation models for scRNA-seq data, for example, by providing a standardized large-scale benchmark dataset and a set of well-tuned reference models.

General cell type classification reflects the ability of models to learn cell types based on transcriptomic profiles, a key abstraction of scRNA-seq data. But, like many supervised machine learning tasks, it is limited by the annotation granularity of the training data. The CELLx-GENE data corpus used here is based on the cell ontology. As the Cell Ontology is still a work in progress, not all cell types can be correctly matched to a corresponding ontology term, this is especially a problem for rare cell types. Moreover, relationships between cell types given by the Cell Ontology are still a topic of active discussion, which can affect the classification metrics discussed here. However, the strength of these current models for automated cell type annotation does not lie in correctly classifying novel cell types, but rather in context-specific suggestions which biologists can further refine. Besides, the models from our paper can be readily retrained once more and better-annotated data becomes available. Related to the previous issue, is the difference in annotation granularity across datasets on CELLxGENE, some authors might annotate a cell as a B cell in their

respective dataset whereas another author might annotate the cell as a naive B cell. This issue is an inherent issue with the CELLxGENE training data, as there is simply information missing in that case. Such an issue cannot be solved by a change in model architecture as this would always mean filling/predicting missing information but needs further data curation efforts to collect the missing information. This, however, would be far beyond the scope of this paper. Nevertheless, despite the potential issues with the training data, our model achieves considerable classification performance (macro F1-Score of 0.83 on our holdout test set). This highlights the point that the issue of missing information is not as pressing as one might think as one can still train a working/useful classifier. We would also like to highlight once more that the current pre-trained scTab models still contain some fairly coarsely annotated cell types despite our efforts to filter out too coarsely annotated cell type labels.

Future work may extend the concept of general cell type classification to less stringent filters of the public data corpus, for example including cells from assay technologies that are not as common as the technologies considered here, and including more refined subtypes of cell types once better-refined training data becomes available at scale. In this context, it becomes particularly interesting to see whether the trend that non-linear models outperform their linear counterparts in distinguishing between fine-grained cell subtypes holds up or even improves. We expect performance for certain tissues to improve along with more terms being added to the Cell Ontology, especially for the study of the brain where the current ontologies lack resolution. Next, we would like to highlight that those efforts will need to take particular care in defining more detailed evaluation metrics, as plain macro F1-scores may not properly reflect the complexity of unbalanced datasets which are not just imbalanced concerning the distribution of cell types but also concerning the distribution of those cell types across e.g. tissues and sequencing protocols. In addition, the performance of machine learning models can be heavily influenced by the composition and quality of the training data. For example, by specifically collecting more training data for cell types or tissues a model struggles with or by being more rigorous with the training data selection through only selecting datasets with high-confidence annotations. Here, we would like to stress that predictions will become more refined, once more refined training data becomes available; and note that the growing magnitude of single-cell data is not limited to transcriptomics; single-cell researchers have increasingly utilized spatial transcriptomics, proteomics, and other multimodal assays to investigate how other features (e.g. chromatin accessibility, DNA methylation, etc.) could be used to distinguish between cell types and cell states. Building on this point a future direction of work would be to extend scTab to take advantage of different input modalities as well once data for those modalities becomes available at an equally large scale, exploiting this feature space to achieve more precise cell type predictions for cell identities. Furthermore, recent efforts to establish foundation models for scRNA-seq data used further tasks to characterize their ability to learn nontrivial representations of cells. We envision further benchmarks to individually address these specific tasks, again focussing on data and strong baseline models. In the context of cellular representation learning, further and more refined choices for data augmentation may be explored. Additionally, these augmentation schemes can then be evaluated in the context of unsupervised representation learning like for example Bootstrap Your Own Latent[45].

Finally, it is critical to make general cell type classification models like scTab easily accessible to the broader community of biological researchers. The Cell Annotation Platform (CAP; https://celltype.info/) has been specifically designed for HCA researchers of all backgrounds to effectively work with the predictions of scTab (as well as other prediction algorithms) directly via their browsers. With the promise of high-confidence predictions of cell types and cell states using a structured vocabulary, as well as the ability to refine and edit these predictions or reannotate cells entirely, researchers will be empowered to annotate their datasets at the current scale required to construct large-scale human cell atlases.

## Methods
### Dataset preparation
The dataset used in this paper is based on the CELLxGENE[15] census version 2023-05-15 (https://chanzuckerberg.github.io/cellxgene-census/index.html). The census version 2023-05-15 is selected as it is a long-term supported (LTS) release and will be hosted by CELLxGENE for at least 5 years. This makes the dataset creation easily reproducible for the foreseeable future. We subsetted to human datasets and used the human protein-coding genes (19,331) as a feature space.

The following criteria are used to filter the human CELLxGENE census data:

1. The census data is subset to primary data only (is_primary_data == True) to prevent label leakage between the train, validation, and test set.
2. Only sequencing data from 10x-based sequencing protocols is used. In terms of the CELLxGENE census, this means subsetting the *assay* metadata column to the following terms: 10×5′ v2, 10×3′ v3, 10×3′ v2, 10×5′ v1, 10×3′ v1, 10×3′ transcription profiling, 10×5′ transcription profiling.
3. The annotated cell type has to be a subtype of the *native cell* label based on the underlying cell type ontology.
4. For each cell type, there have to be at least 5000 unique cells. Otherwise, the whole cell type is dropped from the dataset.
5. Each cell type has to be observed across at least 30 donors to reliably quantify whether the trained classifier can generalize to new unseen donors for each cell type. With the used 70-15-15 train, validation, and test split this means that each cell type is represented with at least 4-5 donors in the validation and test set, respectively.
6. Each cell type needs to have at least seven parent nodes in the cell type ontology. This criterion is used as a heuristic to filter out general cell type labels that do not contain much information.

To be able to better assess how well the trained classifiers generalize to unseen donors or in general to better assess the generalization capabilities of the trained classifiers, the data is split into train, validation, and test sets based on donors and not based on random subsampling. Meaning, each donor is exclusively found either in the training, validation, or test set. Unlike splitting based on e.g. holdout datasets, donor-based splitting mostly preserves the proportion of cells in the training, validation, and test set compared to random subsampling. This is not the case when subsetting the available data based on e.g. datasets, which often results in a very uneven distribution of cells across the training, validation, and test sets as the datasets in the census usually range anywhere between a few thousand cells to a few million cells. Furthermore, dataset-based splitting often makes it hard to ensure that each cell type is observed across both the training data as well as the test data. In the end, the data is split such that 70% of the donors are assigned to the training set and 15% of the donors are assigned to the validation and test set respectively.

The data is size factor normalized to 10,000 counts per cell and log1p-transformed.

The selection described above results in 22,189,056 cells being selected which span 164 unique cell types, 5052 unique donors, and 56 different tissues. Of the 22.2 million cells 15,240,192 cells are assigned to the training set, 3,500,032 are assigned to the validation set and 3,448,832 cells are assigned to the test set.

More detailed explanations and references to the code that can be used to reproduce the above data selection and splitting exactly can be found in the associated GitHub repository under *docs/data.md*.

## Subsampled datasets

We used a subsampled training dataset in the following settings:

Dataset size scaling:

- Random subsampling: 15% subsampling (2.3 million cells), 30% subsampling (4.6 million cells), 50% subsampling (7.6 million cells), 70% subsampling (10.7 million cells), 100% subsampling (15.2 million cells)
- Donor-based subsampling: Subsample to 15% of donors (531 donors / 2.1 million cells), Subsample to 30% of donors (1061 donors / 4.3 million cells), Subsample to 50% of donors (1768 donors / 7.4 million cells), Subsample to 70% of donors (2476 donors / 10.4 million cells), Subsample to 100% of donors (3536 donors / 15.2 million cells)

Data augmentation:

- Subsample to 30% of donors (1061 donors / 4.3 million cells)

In all other cases, the full training dataset is used.

All subsampling is done incrementally, e.g. the 30% subsampled dataset includes all cells/donors that are present in the 15% subsampled dataset and so forth.

## Data loading infrastructure

Training machine learning models on large-scale tabular datasets (which is the case for the scRNA-seq data used in this paper) comes with a set of unique challenges. The first challenge is that the entire dataset does not fit into the memory of a usual server commonly used for training deep learning models. Additionally, the unique nature of tabular data means that you cannot load individual observations from disk efficiently, as individual observations are rather small, and thus loading data points individually creates a lot of random reads which even modern SSDs cannot handle efficiently. Thus, a consecutive block of samples must be loaded at once and then shuffled. Fortunately, there already exist Python libraries that do exactly what is described above. The data loading infrastructure used in this paper is based on the Nvidia Merlin dataloader (https://github.com/NVIDIA-Merlin/dataloader) which gives an easy-to-use API, uses the widely adopted Apache Parquet format to store data on disk and gives performant data loading with GPU-optimized data loaders that directly load the data from disk into GPU memory and then do a 0-copy transfer to PyTorch, TensorFlow or JAX (see Supp. Fig. 8 for details about data loading speed). The above-described data loading infrastructure was fast enough to fully utilize a Nvidia A100 GPU for the models trained in this paper. Moreover, Merlin comes with a wide range of supporting infrastructure like Docker containers (https://catalog.ngc.nvidia.com/orgs/nvidia/teams/merlin/containers/merlin-pytorch) from the NGC container hub which makes it easy for people to start using Merlin without the need to set up Python environments first.

## Data augmentation

The idea behind the data augmentation strategy developed in this paper is that the difference in raw gene space between the same cell type observed across two donors can be used as a data augmentation vector that can simulate how the gene expression of a cell might look like for a different donor. The general idea behind data augmentation is to have easy-to-compute transformations that can be applied during model training. Thus, in this case, we pre-compute augmentation vectors that can be added to the observed gene expression of a cell to artificially increase the training data size during model training:

$$x_{augmented} = x_{original\ cell} \pm x_{augmentation\ vector} \quad (1)$$

**Calculation of augmentation vectors.** The augmentation vectors are calculated as follows:

1. Subsample 500,000 cells from the training data to have an even distribution across cell types.
2. Calculate the mean centroids grouped by cell type and donor
3. Calculate the difference vectors between the mean centroids from step 2 by cell type.
4. Set all values in the range [−0.25, 0.25] to zero to enforce more sparse augmentation vectors.
5. Clamp the resulting augmentation vectors to the interval of [−1.5, 1.5] to remove outlier values.
6. Filter the resulting augmentation vectors for outliers by only sampling the used augmentation vectors from the most prominent k-means clusters (clustering is done with 50 clusters) → sample e.g. 5000 augmentation vectors from the biggest k-means clusters (clusters with more than 2000 difference vectors).

Note that Step 6 is used to enforce the selection of cell type independent augmentation vectors. As can be seen in Supp. Fig. 9a, some of the calculated augmentation vectors are influenced by the cell type based on which they are calculated. This can be problematic if the augmentation vectors are applied in a cell type-independent fashion e.g. by randomly sampling augmentation vectors. To ensure that the augmentation vectors can be applied in cell type independent fashion, we filtered the augmentation vectors in Step 6 to only select augmentation vectors that are mostly cell type independent. This filtering based on K-means clustering indeed results in mostly cell type independent augmentation vectors (Supp. Fig. 9b).

The associated parameters of our data augmentation strategy should be tuned in a similar fashion as one would tune the hyperparameter of a neural network. This means calculating the augmentation vectors based on the training split, then selecting the best parameter set based on the validation split, and finally, reporting the performance on the holdout test set.

**Calculation of augmented gene expression vectors during model training.** The augmented gene expression vectors are calculated as follows:

1. Sample an augmentation vector $x_{augmentation\ vector}$ from the set of augmentation vectors
2. Sample whether the augmentation vector is added to or subtracted from the original gene expression vector $x_{original\ cell}$
3. Add/subtract the sampled augmentation vector to the original gene expression vector and clamp all values of the newly created vector to be within the interval of [0., 9.]

**Explained variance by cell type before and after data augmentation.** To estimate how our data augmentation influences the proportion of the overall variance that can be attributed to cell type variation, we fitted a linear regression (sci-kit learn LinearRegression) model which predicts the normalized gene expression based on the cell type and donor of each cell. This corresponds to the following design matrix:

$$\hat{y} = 1 + onehot(celltype) + onehot(donor) \quad (2)$$

In the next step, the $R^2$ score of the model fitted on the original/non-augmented data is compared to the one from the model fitted on the augmented data to show how the amount of total variation in gene expression, which can be attributed to the cell type, changes.

## Ontology-corrected cell type classification

The classification performance of the trained models in this paper is evaluated based on the macro average of the F1-scores for each individual cell type. The macro average is used to give each cell type the same weight in the overall classification performance. The F1-score is

calculated as follows:

$$F1 - score = 2 \frac{precision \cdot recall}{precision + recall}$$

$$= \frac{2 \cdot tp}{2 \cdot tp + fp + fn} \quad (tp : \text{true positives}, fp : \text{false positives}, fn : \text{false negatives})$$

$$(3)$$

In order to deal with the often different granularity of annotations (e.g. label T-cell vs label CD4-positive, alpha-beta T cell) the following rules are applied to evaluate whether a prediction is considered right or wrong. A prediction is considered as right, either if the classifier predicts the same label as supplied by the original dataset, or if the classifier predicts a subtype of the label provided by the original dataset - we consider this as a right prediction as the prediction agrees with the true label up to the annotation granularity the author provided. The subtype relations are evaluated based on the Cell Ontology[31]. An example is if the model predicts the label CD4-positive, alpha-beta T cell when the author annotated cell type is T cell. Moreover, a prediction is considered wrong if the classifier predicts a parent cell type of the true label - we consider this as a wrong prediction as the author supplied a more fine-grained label that the classifier should replicate. An example is if the classifier predicts the label T cell while the cell is labeled as a CD4-positive, alpha-beta T cell in the original dataset. In all other cases, the prediction is considered wrong. Furthermore, the lookup of child nodes in the cell ontology is based on the Ontology Lookup Service (OLS): https://www.ebi.ac.uk/ols/ontologies/cl [31].

## Performance evaluation on coarse cell type labels

To give an impression of how scTab performs on more coarse cell type labels, we evaluated the performance of our scTab model on a set of more coarsely annotated cell type labels. We selected coarse cell type labels based on the information content score provided by the cell ontology (https://github.com/INCATools/ubergraph?tab=readme-ov-file#graph-organization). The information content score is calculated based on the count of terms related to a given cell ontology term and is in the interval [0, 100], where 100 corresponds to a very specific term with no subclasses. Based on the information content score we used the following rules to define a set of coarse cell type labels:

1. Get all cell type labels present in the CELLxGENE census which are a subset of the native cell cell type label.
2. Keep all cell type labels with an information content score of less or equal to 60.
3. Assign each cell type label to one of the coarse cell type labels from step 2. If based on the cell type ontology, a cell type label can be assigned to more than one of the coarse labels, we only assign it to the coarse label with the highest information content score. Example: the label alpha-beta T cell would be assigned to T cell as the coarse label and not lymphocyte.
4. Use the grouping from Step 3 to assign each of the fine-grained cell type labels to a coarse cell type label.

Moreover, we would like to note that we did not retrain the model from scratch for the evaluation on the coarse cell type labels. Instead, we aggregated the predictions of the model that was trained on the fine-grained cell type labels. For instance, all predictions of mature T cell subtypes count as predicting the label mature T cell (based on the underlying hierarchy of the Cell Ontology).

## Model details

**scTab model.** Our implementation of scTab is based on the TabNet architecture[33] and is mostly taken from the dreamquark-ai/tabnet GitHub repository with some adaptation towards the single-cell use case. The input to the model is all 19,331 protein-coding genes (GENCODE v38/Ensembl 104) selected from the CELLxGENE census data. Moreover, unlike in the original TabNet model, we normalized the input data before feeding it into the neural network. scRNA-seq data is often normalized to have 10,000 counts per cell and is then log1p transformed afterward[6,12,22], we applied the same normalization for our scTab model on top of the simple batch normalization layer, which is used in the original TabNet model to normalize the input features, as such a non-linear normalization cannot be achieved by a simple batch normalization layer.

The adapted TabNet architecture for scTab (Fig. 1b) consists of two key building blocks: The first building block is the feature transformer, which is a multi-layer perceptron with batch normalization (BN), skip connections, and a gated linear unit nonlinearity (GLU). The feature transformer maps from the input gene expression space to an n_d + n_a dimensional latent space. In the next step, the n_d + n_a dimensional embedding is split into two parts: one with dimension n_d and one with dimension n_a. The part with dimension n_d is used to classify the different cell types and the second part with dimension n_a is used to calculate the attention masks. The feature attention mask is obtained by using a single linear layer followed by a batch normalization layer that maps from the feature attention embedding to the input feature space. The feature attention mask is then obtained by applying the 1.5-entmax[46] function to the output of the linear projection layer. Using the 1.5-entmax function instead of the sparsemax function, which is used in the original TabNet model, improved training dynamics and yielded slightly higher model performance. The 1.5-entmax function is defined as follows:

$$H_{1.5}^T(p) = \frac{1}{1.5 \cdot (1.5-1)} \sum_j \quad (p_j - p_j^{1.5}) \text{ for any } p \in \Delta^d \quad (4)$$

$$1.5entmax(z) = argmax_{p \in \Delta^d} \, p^T \cdot z + H_{1.5}^T(p) \quad (5)$$

After obtaining the feature attention mask, the masked input features are fed into the feature transformer to obtain the feature embedding used to classify cell types. Thus, by giving the neural network the ability to mask individual input features, it can focus its network capacity only on more reliable input features. In contrast to the original TabNet model, we only used a single decision step as using more than one decision step only yielded marginal performance improvements and did not justify the increased computational costs.

The objective function used to train scTab is a cross-entropy loss where each cell type label is weighted in correspondence to its relative frequency in the training data to account for the strong class imbalance in the training data:

$$weight_{celltype} = \frac{n_{samples}}{n_{classes} \cdot \Sigma_{cell \, in \, cells} \quad label_{cell} == celltype} \quad (6)$$

The models for Fig. 1 and Fig. 3 were fitted with our proposed data augmentation strategy. The models for Fig. 2 were fitted without data augmentation to better show the scaling with respect to the training data size.

List of used hyperparameters:

| Parameter | Value |
| --- | --- |
| batch_size | 2048 |
| learning_rate | 0.005 |
| learning rate scheduler | torch.optim.lr_scheduler.StepLR gamma = 0.9 step_size = 1 epoch |
| optimizer | torch.optim.AdamW |
| weight_decay | 0.05 |
| n_d | 128 |

| | |
|---|---|
| n_a | 64 |
| n_shared | 3 |
| n_independent | 5 |
| n_steps | 1 |
| lambda_sparse | 1e-5 |
| mask_type | entmax |
| virtual_batch_size | 256 |
| augment_training_data | True |

**XGBoost model.** The input to the XGBoost model is a 256-dimensional PCA embedding due to the high memory usage and runtime of the XGBoost model. The PCA is only fitted on the training data to have a clear separation between the training and test set. Furthermore, the data is normalized to 10,000 counts per cell and is then log1p-transformed before calculating the PCA embeddings. The XGBoost model is fitted with the multi:softprob objective function and like for the scTab model classes are weighted in accordance to their relative frequency in the training data.

List of non-default hyperparameters:

| Parameter | Value |
|---|---|
| n_estimators | 800 |
| eta | 0.05 |
| subsample | 0.75 |
| max_depth | 10 |
| early_stopping_rounds | 10 |

For the benchmarks in this paper, we used XGBoost version 1.6.2

**Multi-layer perceptron model (MLP).** The input to the model is all 19,331 protein-coding human genes selected from the CELLxGENE census data. The model is trained to predict the corresponding cell type label for each cell with a cross-entropy loss where each cell type is weighted in correspondence to its relative frequency (see scTab model).

The input count data is normalized to 10,000 counts per cell and is then log1p-transformed before feeding it into the model.

List of used hyperparameters:

| Parameter | Value |
|---|---|
| batch_size | 2048 |
| learning_rate | 0.002 |
| learning rate scheduler | torch.optim.lr_scheduler.StepLR gamma = 0.9 step_size = 1 epoch |
| optimizer | torch.optim.AdamW |
| weight_decay | 0.05 |
| n_hidden | 8 |
| hidden_size | 128 |
| dropout | 0.1 |
| augment_training_data | True |

**Optimized linear model.** The input to the model is all 19,331 protein-coding human genes selected from the CELLxGENE census data. The model consists of a single weight matrix and bias vector and is trained to predict the corresponding cell type label for each cell with a cross-

entropy loss where each cell type is weighted in correspondence to its relative frequency (see scTab model).

The input count data is normalized to 10,000 counts per cell and is then log1p transformed before feeding them into the model.

List of used hyperparameters:

| Parameter | Value |
|---|---|
| batch_size | 2048 |
| learning_rate | 0.0005 |
| learning rate scheduler | torch.optim.lr_scheduler.StepLR gamma = 0.9 step_size = 1 epoch |
| optimizer | torch.optim.AdamW |
| weight_decay | 0.01 |

**CellTypist model.** The CellTypist[6] model was fitted in accordance with the best practice tutorial supplied on the CellTypist website with the difference that the mean centering step was disabled (with_mean=False) as this negatively impacted model performance and increased memory usage. Furthermore, the training data was sub-sampled to 1.5 million cells to keep both the memory usage (350GB of max memory) and runtime in check.

List of non-default hyperparameters:

| Parameter | Value |
|---|---|
| feature_selection | True |
| use_SGD | True |
| mini_batch | True |
| batch_number | 1500 |
| epochs | 10 |
| with_mean | False |

For the benchmarks in this paper, we used CellTypist version 1.5.3

**scGPT (zero-shot setting).** We evaluated the performance of scGPT in the zero-shot setting, meaning we used the pre-trained whole-human scGPT model to get cell embeddings and used those embeddings as input to a logistic regression classifier. The logistic regression classifier was trained on a random subsample of 1,500,000 cells from the training data.

List of non-default hyperparameters for cuml LogisticRegression:

| Parameter | Value |
|---|---|
| class_weight | balanced |
| max_iter | 5000 |
| C | 1000 |

For the benchmarks in this paper, we used cuml version 23.10 and scgpt version 0.1.7.

Whole-human scGPT model: https://drive.google.com/drive/folders/1oWh_-ZRdhtoGQ2Fw24HP41FgLoomVo-y?usp=sharing.

**scGPT (fine-tuned).** We fine-tuned the scGPT in accordance with the following example notebook provided by the authors of the scGPT paper: https://scgpt.readthedocs.io/en/latest/tutorial_annotation.html

Due to the high memory usage of scGPT, we were only able to fine-tune the scGPT model on a random subsample of 150,000 cells of our training data. For our benchmark, we fine-tuned the whole-human

scGPT model: https://drive.google.com/drive/folders/1oWh_-ZRdhtoGQ2Fw24HP41FgLoomVo-y?usp=sharing.

**Universal Cell Embedding (UCE) (zero-shot).** As the UCE model[47] is very resource intensive - even when just using it for inference, we used the pre-computed UCE embeddings which are hosted by CELLxGENE (https://cellxgene.cziscience.com/census-models) and evaluated them in the linear probing setting. This means fitting a logistic regression model based on the embeddings obtained by UCE. Unfortunately, these pre-computed embeddings only exist for census version 2023-12-15 which is missing some datasets that were included in census version 2023-05-15 (the census version used in this paper). In numbers, this means, we could only evaluate the UCE model on 736 of the 758 donors from our test data.

We fitted the logistic regression classifier on a random subsample of 1,500,000 cells of the training data and then evaluated this classifier on the reduced test data (only 736 of the original 758 donors from our test data).

### Uncertainty quantification for scTab model

The uncertainty quantification for scTab is based on deep ensembles[34] using $1 - maximum\ predicted\ probability$ as an estimate for the model uncertainty. Deep ensembles are commonly used to assess the uncertainty in predictions of neural networks. They are simple, yet achieve state-of-the-art results: one just averages the predicted probabilities across several networks that were independently trained (each with a different random initialization of the weights). In our case, we averaged the predictions across 5 models.

To assess how well one can identify cell types that are not present in the training data or cells with wrong predictions we split the CELLxGENE data into three parts:

- **Group 1: Correct Predictions** Cell types that are present in the training data and which are predicted correctly by the model (this serves as a reference group). This group is referenced as in-distribution (right prediction) or simply as Group 1 below.
- **Group 2: Incorrect Predictions** Cell types that are present in the training data but which the model predicted wrongly to assess how well wrong predictions can be distinguished from right predictions based on the uncertainty scores. This group is referenced as in-distribution (wrong prediction) or simply as Group 2 below.
- **Group 3: Absent in Training Data** Cell types that are not present in the training data to assess how well unknown cell types can be identified based on the uncertainty scores provided by scTab. These are the cell types that we excluded from the CELLxGENE training data because there were too few observations present. This group is referenced as out-of-distribution or simply as Group 3 below.

Now, to understand the quality of uncertainty estimates, we want to assess how well Group 2 and Group 3 (incorrect and absent) can be separated from reference Group 1 (correct). Note that the separation between Group 1 and Group 2 (correct vs incorrect) measures how well the uncertainty scores can be used to assess whether a model prediction can be trusted or not, and the separation between Group 1 and Group 3 (correct vs absent) gives an estimate of how well the uncertainty scores can be used to detect new/unseen cell types. The model uncertainty is defined as follows (logits are the outputs of the last layer of the neural network):

$$uncertainty = 1. - \max(p) \tag{7}$$

$$p = softmax(logits) \tag{8}$$

To get a first impression, one can look at the distribution of uncertainty scores conditioned on which one of the three groups a cell

belongs to (see Supp. Fig. 4c). As expected, one can see that the uncertainty scores for Group 2 and Group 3 are usually a lot higher than for Group 1.

Now, to provide a more mathematically rigorous benchmark, one can measure how well one can distinguish Group 2 and Group 3 from the reference Group 1 based on the uncertainty scores provided by the scTab model by looking at the area under the curve of the receiver operating characteristic (ROC-AUC). A ROC-AUC score of 1.0 means that the groups can be perfectly separated and a score of 0.5 means that there is no separation between the groups based on the uncertainty scores (see Supp. Fig. 4d). The above approach can also be used to assess how the quality of uncertainty estimates improves with the number of models in the deep ensemble. Looking at the results, one can see that for both cases our uncertainty estimates provide a useful way to distinguish between the groups. Group 1 and Group 3 can be separated with an ROC-AUC score of 0.782 and Group 1 and Group 2 can even be separated with an ROC-AUC score of 0.891.

In practice, biologists and computational biologists can overlay the uncertainty estimates from scTab alongside the predicted cell type labels and their defined clustering on a UMAP or tSNE visualization of their data to hint at which predictions are associated with higher uncertainty and, hence, should be investigated in more detail (see Supp. Fig. 4a).

### Statistics and reproducibility

No statistical method was used to predetermine sample size. We simply used all the available data from the CELLxGENE data corpus (version 2023-05-15) subject to the our filtering criterion described in the Dataset preparation section (Methods).

### Reporting summary

Further information on research design is available in the Nature Portfolio Reporting Summary linked to this article.

## Data availability

All relevant data supporting the key findings of this study are available within the article and its Supplementary Information files. All data used in this manuscript can be downloaded from CELLxGENE[15] (census version "2023-05-15"). Moreover, for ease of use, the processed data and checkpoints of the trained models can be download here: Data: https://pklab.med.harvard.edu/felix/data/merlin_cxg_2023_05_15_sf-log1p.tar.gz (164GB) Checkpoints: https://pklab.med.harvard.edu/felix/data/scTab-checkpoints.tar.gz (8.1GB) Source data are provided with this paper.

## Code availability

GitHub - All code: https://github.com/theislab/scTab/tree/devel[48] GitHub - Tutorials: Data loading tutorial: https://github.com/theislab/scTab/blob/devel/notebooks-tutorials/data_loading.ipynb. Loading pre-trained models: https://github.com/theislab/scTab/blob/devel/notebooks-tutorials/model_inference.ipynb.

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

## Acknowledgements

We thank the Cell Annotation Platform (CAP) team - especially Uğur Bayındır - for their continuous support and feedback in developing and

evaluating scTab, as well as Peter Kharchenko for his guidance on the implementation of new methods and computational frameworks on CAP. We particularly wish to thank David Osumi-Sutherland for discussions regarding how to best utilize the Cell Ontology for analyses during manuscript revisions. We thank Giovanni Palla, Luke Zappia, Alejandro Tejada Lapuerta, and Lukas Heumos for their suggestions that made the manuscript stronger. The authors also gratefully acknowledge the computational and data resources provided by the Leibniz Supercomputing Centre (www.lrz.de) and the system storage provided by TUM-DSS. The authors would also like to thank Keith Bayer and the Web and Advanced Research Platforms at Harvard Medical School for their support throughout this project. D.S.F. acknowledges support from a German Research (DFG) fellowship through the Graduate School of Quantitative Biosciences Munich (QBM) [GSC 1006 to D.S.F.] and by the Joachim Herz Foundation. This work was supported in part by funding from the Eric and Wendy Schmidt Center at the Broad Institute of MIT and Harvard. A.C.V. acknowledges support from the National Institute of Health (DP2CA247831). This work was supported in part by funding from Schmidt Futures for the Cell Annotation Platform (to A.C.V. and F.J.T), as well as by the Helmholtz Association's Initiative and Networking Fund on the HAICORE@FZJ partition. This publication is part of the Human Cell Atlas – www.humancellatlas.org/publications

## Author contributions

F.F., D.S.F., and F.J.T. worked on pilot analyses and conceived the project. F.F. conducted the model implementation and analyses with input from E.B., R.M., A.I., D.S.F., and F.J.T. F.F. wrote and tested the software with E.B., R.M., and A.I. A.C.V. oversees and supports the Cell Annotation Platform effort. F.F., E.B., D.S.F., and F.J.T. wrote the manuscript. All authors discussed the results and commented on the manuscript.

## Funding

## Competing interests

F.F. consults for Dermagnostix GmbH. F.J.T. consults for Immunai Inc., Singularity Bio B.V., CytoReason Ltd, Cellarity, and Curie Bio Operations, LLC, and has an ownership interest in Dermagnostix GmbH and Cellarity. The remaining authors declare no competing interests.
