## [Peer Review File · Nature Communications]

scTab: Scaling cross-tissue single-cell annotation modelsReviewer #1 (Remarks to the Author):

Summary of the study

Cell type annotation remains a time-consuming and often subjective task in single-cell transcriptomics data analysis. Several automated annotation approaches have been published in recent years; however, they usually only encompass either a specific tissue or cellular compartment and are therefore not broadly applicable across organs and cell lineages. In this manuscript, Fischer et al. present an innovative approach for automated cell type annotation across tissues and cell lineages using a deep learning framework and a vast reference training dataset. The method, called scTab, is based on an attention-based model tailored to the structure and properties of single-cell transcriptomics data. The authors used a training dataset from the CELLXGENE repository with around 22 million human cells and 164 different cell types. Moreover, they used data augmentation to further boost model generalization. Finally, the authors performed a comprehensive benchmarking against alternative linear and non-linear models and show that scTab exhibits higher performance when compared to community standards. This new cell annotation approach has the potential to set a new standard for the single-cell community, generating more complex models, with larger and more diverse training datasets. We consider this work of great value for the single-cell community. We have summarized below the key points that we consider are necessary to address:

Major points:

- 1) In Fig. 1c, the authors show that scTab outperformed the rest of models benchmarked. In order to bring across the specific impact of this higher performance, could the authors provide concrete examples of how label prediction improved the annotation of specific cell types?
- 2) Following from the previous point, the authors explain how they have considered ontology during the preparation of the training set. The level of ontology hierarchy or granularity of cell type annotation is known to affect the performance of cell type classifiers (see as an example the two levels of hierarchy in CellTypist "Immune_All" models). Could the authors expand on how is performance of scTab influenced by the level of granularity in cell type annotations?
- 3) In lines 222-224, the authors state: "Furthermore, one can see that classes on which errors were made tended to be those that were represented by few observations (Fig. 1g)." Interestingly, the scatterplot in Fig. 1g also shows that a group of cell types with low cell counts show F1-scores as high as more abundant cell type classes. Can the authors comment on their interpretation of this phenomenon and whether this affects specific cell lineages or higher levels of cell type ontology terms?
- 4) Could the authors show a runtime comparison of the training with the models compared in Fig. 1c?
- 5) In Fig 2e, the authors show the comparison in performance in cell classification of organ-specific training and cross-tissue training with lung data, indicating that "cross-organ cell type classification benefits from using non-linear models". To make this statement, could this comparison be shown for all individual tissues or, at least, a higher number of per-tissue models?
- 6) In connection with the previous point, could the authors show whether the improvement between cross-organ and individual tissue models is driven by specific cell type classes? Do the classes driving the improvement in performance change according to tissue? Are misclassified cells always the same cell type? The authors could provide a scatter plot showing the correlation between per-class F1-scores in scTab versus the optimized linear model and show the actual cell type labels that vary the most between models.
- 7) A more detailed explanation of common use cases for the scTab pre-trained models in different settings would be advisable given the broad interest in the single-cell community. Also, an indication of use cases for newly trained models.

8) The authors made available both the training data as well as the code for generating the scTab models at <https://github.com/theislab/scTab>. It is however unclear if the authors intend their pre-trained models to be used by the community. If so, instructions on how to use the scTab models for prediction on new datasets should ideally be included as part of the documentation.

9) Only 10X Genomics data is used for training the models. Could the authors comment on the rationale for this and could they provide insights into performance across chemistries given that CELLXGENE already contains non 10X Genomics datasets?

10) Universal Cell Embeddings (UCE) has been recently presented as a universal representation of all cells in the human body that can be used for cell type annotation (DOI: <https://doi.org/10.1101/2023.11.28.568918>). Could the authors comment on how this publication relates to their work and whether it would be appropriate to include UCE in their benchmarking?

Minor points:

1. We would recommend an alternative representation for Fig1a. Most of the cell labels are not readable and the relative contribution of cell types and tissues is not clear.

2. We would suggest modifying "Linear (ours)" by for example "Optimized linear"

3. We would suggest rewriting the following sentence in lines 89-91 as it is not understandable: "However, those efforts on foundation models benchmark cell representations on diverse tasks, so far without context on deep models specifically designed for cell type annotation."

4. In lines 98-100, the "zero-shot" concept appears for the first time, and it is mentioned several times throughout the manuscript. We suggest including a brief definition of the term: "Furthermore, according to recent benchmarks^{25,26}, recent foundation models often only show comparable performance to simpler and often linear reference models in the zero or few-shot setting."

5. We suggest omitting the word "significant" in the following sentences:

Lines 196-199: "We implemented a logistic regression-based model not subject to these limitations (Methods) and trained this model with a cross-entropy loss. This model significantly outperformed the CellTypist reference model and achieved a macro F1-score of 0.7848 ± 0.0001 (Fig. 1c)".

Lines 204-206: "The nonlinear models significantly outperformed the linear model (0.8295 ± 0.0007 macro F1-score for scTab (fitted with data augmentation), 0.8127 ± 0.0005 for XGBoost, 0.7971 ± 0.0012 for MLP (fitted with data augmentation)) (Fig. 1c, Supp. Table 1)".

6. If Fig. 2b, neither the upper title: "number of unique donors" nor the lower title: "number of training samples", seem to indicate the number of cells sampled. Did the authors intend to indicate the "number of cells in the training set" in the lower title?

7. Lines 63-65: We would suggest omitting the word "surprisingly":

"However, cross-tissue classifiers trained on large-scale data collections that annotate cells from heterogeneous sources, irrespective of tissue of origin and assay type, are surprisingly slow to emerge."

8. Reference 19 has been cited as a preprint but is now published in Nature methods.

Reviewer #2 (Remarks to the Author):

I co-reviewed this manuscript with one of the reviewers who provided the listed reports as part of the Nature Communications initiative to facilitate training in peer review and appropriate recognition for co-reviewers.

Reviewer #3 (Remarks to the Author):

The authors have developed a deep-learning-based classification algorithm for annotating cell-type information, leveraging large-scale reference datasets. This method, adapted from the TabNet architecture, was originally designed for tabular data with adaptation made for its application to scRNA-seq data. scTab integrates feature and attentional transformer blocks for feature selection.

While the manuscript is generally well-written and engaging, I have some reservations about the novelty, significance, and generalizability of the proposed methods. A more in-depth discussion comparing scTab with existing methodologies in this field would be needed to enhance the manuscript's contribution to the scientific community. Additionally, the choice of evaluation metrics for scTab warrants further elaboration. The details of my comments are below:

Major Comments:

1. The methods presented in the manuscript extend from currently available methods primarily designed for tabular data. I appreciate the authors' efforts in adapting these methods to the scRNA-seq context, particularly in terms of handling large-scale datasets. However, there are several key challenges that seem to remain unaddressed, which affects the perceived novelty and interest in the methods. For example:

(Line 108) The author discusses the potential overlap between labels (e.g., 'naive B cell' and 'B cell' in Figure 1), a common issue in real datasets. The approach to address this involves adopting CELLXGENE as a root dataset. However, it appears that scTab does not fundamentally address the hierarchical structure of the labeling inherent to such datasets.

2. One of the main contributions of scTab is its generalizability, which is facilitated by the heterogeneity of cells provided by the large-scale training corpus. However, I believe its generalizability has not yet achieved its maximum potential, and several improvements are needed.

(Line 153) The authors limit the model to a single genomic sequencing platform (10X Genomics). However, it's unclear how the method would perform with query data from different or mixed platforms. Can the authors clarify if the method would still be effective when the training and testing data originate from diverse sequencing platforms? It would be beneficial to conduct experiments to assess how discrepancies in data platforms might influence performance.

3. The scTab is designed to focus on cross-tissue classification. However, I am curious about how the method addresses batch effects or differences arising from each tissue, beyond simply merging the datasets. Could the authors elaborate on any specific strategies or techniques employed by scTab to mitigate batch effects inherent in cross-tissue analyses?

4. (Line 156) The authors have set a threshold to remove rare cells with fewer than 5000 instances. However, I believe this threshold may be too stringent. In the context of small-scale supervised learning, 5000 instances represent a significant amount, and this criterion could potentially exclude relevant data.

5. How does the model handle new cell types in the query data that are not present in the reference dataset? This aspect is crucial for assessing the model's applicability to diverse and evolving datasets.

6. The author describes a data augmentation procedure involving the shifting of measurements across donors, learned from other cell types. For instance, the augmentation vector for T cells is captured using B cells. However, I am concerned about the validity of this approach. The gene space might interact differently with various cell types across donors, potentially affecting the reliability of the augmentation (because the augmenting vector is influenced by cell type). Could the authors clarify how this method accounts for the potential interaction between the cell types and donors in the change of gene space?

7. (Line 193) While I appreciate the presentation of the standard deviation in the results, the method of its calculation remains unclear. The small magnitude of the standard deviation values is particularly surprising. Could the authors provide more details on how these values were computed? Specifically, does the repetition of the experiment involve bootstrapping of the sample?

8. (Lines 205-210) The comparisons made with existing methods appear to be potentially unfair. For example, the XGBoost model was not fitted with data augmentation, unlike the proposed method. Similarly, scGPT was used without fine-tuning. These discrepancies might lead to a skewed assessment of the performance of the proposed method against these established techniques.

9. (Line 269) Could the authors provide the p-value associated with the reported correlation of -0.55?

10. (Line 562) The authors have chosen macro F1-score as an evaluation metric, which, as noted in the discussion, could be influenced by class size imbalance. It may be beneficial for the authors to also consider presenting the micro F1-score.

11. (line 591) This sentence is confusing. A revision might be needed: " On the one hand, the feature transformer is used to get a lower dimensional representation (described by dimensionality n_d) which is used to classify cell types. On the other hand, it is used to get a lower dimensional representation (described by dimensionality n_a) from which the feature attention mask is calculated. "

Minor Comments:

12. (line 357) Typo "datase"

13. (line 600) Notation "p" and " p_j " are undefined.

Reviewer #4 (Remarks to the Author):

This study establishes a comprehensive dataset designed to facilitate the training and evaluation of a cross-tissue cell type classification model. This initiative represents a substantial advancement in the landscape of cell type identification, making noteworthy contributions to the field. Additionally, the paper introduces scTab, an automated cell type prediction model meticulously tailored for tabular data, employing a feature-attention-based methodology. The model undergoes training using an innovative data augmentation scheme applied across an extensive repository of single-cell RNA-seq observations, encompassing a total of 22.2 million human cells. In summary, this paper holds substantial potential for impact, supported by its anticipated high citation rate. The overall presentation of the paper is commendable and warrants publication with only minor revisions.

Several queries arising from the content have proven perplexing, and your insights would be greatly appreciated.

(1) I suggest using specific model names to differentiate between the "linear (ours) model" and a generic "linear model."

(2) Could you please provide a clear definition for the "cross-tissue annotation task"? Initially, I inferred that the model was trained on data from one tissue and tested on another. However, upon careful examination of the paper, it became evident that this interpretation was not accurate.

(3) Clarification is sought regarding any distinctions between "cross-tissue annotation" and "general cell type annotation."

(4) The decision to reject sequence-based models, such as Transformers, due to the unordered nature of gene expression profiles is reasonable. Instead, the authors opted for a recent architecture explicitly designed for tabular data, namely TabNet. Additionally, scGPT incorporates a novel attention mechanism to address this challenge. While it is conceptually reasonable that sequence models may not be ideal for non-sequential data, uncertainty remains regarding whether the unordered nature of gene expression profiles genuinely impacts the performance of Transformer-based models. Some Transformer-based models employed for cell type identification, such as CIFORM[1] and TOSICA[2], despite overlooking the unordered nature of genes, have demonstrated commendable performance. I recommend comparing this aspect with TOSICA and CIFORM. If a direct comparison is considered inappropriate, please provide a suitable rationale.

(5) The paper introduces scTab as a tool for cell type annotation. I am intrigued by its performance on widely used benchmarking datasets, such as the Zhang68K[3] dataset.

References:

1. Xu J, Zhang A, Liu F et al. CIFORM as a Transformer-based model for cell-type annotation of large-scale single-cell RNA-seq data, *Brief Bioinform* 2023;bbad195.
2. Chen J, Xu H, Tao W et al. Transformer for one stop interpretable cell type annotation, *Nat Commun* 2023;14:223.
3. Zheng GX, Terry JM, Belgrader P et al. Massively parallel digital transcriptional profiling of single cells, *Nat Commun* 2017;8:14049.

**Reviewer #1**

**Cell type annotation remains a time-consuming and often subjective task in single-cell**
**transcriptomics data analysis. Several automated annotation approaches have been**
**published in recent years; however, they usually only encompass either a specific tissue or**
**cellular compartment and are therefore not broadly applicable across organs and cell**
**lineages. In this manuscript, Fischer et al. present an innovative approach for automated**
**cell type annotation across tissues and cell lineages using a deep learning framework and a**
**vast reference training dataset. The method, called scTab, is based on an attention-based**
**model tailored to the structure and properties of single-cell transcriptomics data. The**
**authors used a training dataset from the CELLxGENE repository with around 22 million**
**human cells and 164 different cell types. Moreover, they used data augmentation to further**
**boost model generalization. Finally, the authors performed a comprehensive benchmarking**
**against alternative linear and non-linear models and show that scTab exhibits higher**
**performance when compared to community standards. This new cell annotation approach**
**has the potential to set a new standard for the single-cell community, generating more**
**complex models, with larger and more diverse training datasets. We consider this work of**
**great value for the single-cell community. We have summarized below the key points that**
**we consider are necessary to address:**

We thank the reviewer for the accurate summary and kind comments as well as constructive
suggestions!

**Major points:**

**1. In Fig. 1c, the authors show that scTab outperformed the rest of models benchmarked. In**
**order to bring across the specific impact of this higher performance, could the authors**
**provide concrete examples of how label prediction improved the annotation of specific cell**
**types?**

**Author response:**

Thank you for this question. We agree that this should be better clarified; indeed, explicitly detailing
this in the main manuscript would support our argument that this method provides novel utility for
practicing biologists.

When looking at the per cell type F1-Scores (see barplot below or Supp. Fig. 7a - the barplot is
split across two rows to make the cell type labels easier to read), one can see that scTab generally
performs at least equally well as the optimized linear model (except for very few exceptions),
especially for broad classes of cell types encoded in the Cell Ontology. This is arguably not too
surprising, given broad classes of cell types should be far easier to classify by the very nature of
being a broad category encompassing many refined cell types. However, when looking more
closely, one can see that for some cell types scTab outperforms the optimized linear model by a
considerable margin, in particular for distinguishing between closely related classes of refined cell
types.

To investigate this more deeply, we highlighted the top 25 cell types with the biggest difference in

classification performance (measured by F1-Score) between our scTab model and the optimized

linear model in the barplot below (or Supp. Fig. 7b).

Looking at the barplot in more detail and especially at the cell type labels, one can see that scTab
 does particularly well in distinguishing fine-grained cell types from each other — most prominently
 when distinguishing between T cell subtypes. The Cell Ontology is especially detailed for T cell
 subtypes, making it possible to resolve fine-grained differences between subtypes. This can be
 seen well by looking at the overall fraction of T cell subtypes in the overall training data from
 CELLxGENE, where T cell subtypes make up about 23% of the 164 unique cell type labels. To
 assess this trend quantitatively, we plotted on the one hand the classification performance of
 distinguishing T cells in general from other cell types and on the other hand for correctly
 distinguishing T cell subtypes in the barplot below (or Supp. Fig. 8).

 One can see that both scTab and the optimized linear model do well in distinguishing T cells from
 other cell types, where both models achieve almost perfect classification performance. However,
 one can see the potential of our non-linear scTab model when distinguishing T cell subtypes where
 scTab gives an average improvement in macro F1-Score of 0.103 across all T cell subtypes
 (compared to only 0.045 across all cell types).
 Furthermore, when grouping the cell types into bins based on the difference in F1-score between
 scTab and the optimized linear model, one can see that the fraction of T cell subtypes in each bin
 increases the higher the difference in F1-score between the two models gets (see barplot below):

 In summary, when looking at how well our scTab model and the optimized linear model can
 distinguish T cell subtypes, one can see the potential of our non-linear scTab model for
 distinguishing closely related cell ontology terms from each other.

 We've added the text below to the manuscript to summarize the findings above:

 Now, looking in more detail where the performance improvements of our scTab over the
 optimized linear model stem from, one can observe that scTab performs particularly well in
 distinguishing between closely related classes of refined cell types - most prominently when
 distinguishing between T cell subtypes (Supp. Fig. 7b). The Cell Ontology is especially

detailed for T cell subtypes, making it possible to resolve fine-grained differences between
subtypes. To assess the above trend quantitatively, we, on the one hand, looked at the
classification performance of distinguishing T cells in general from other cell types and on
the other hand for correctly distinguishing T cell subtypes for scTab and the optimized linear
reference model: Both scTab and the optimized linear model do well in distinguishing T
cells from other cell types, where both models achieve almost perfect classification
performance (F1-score of 0.98 for scTab and 0.97 for the optimized linear model). However,
one can see the potential of our non-linear scTab model when distinguishing T cell
subtypes where scTab gives an average improvement in macro F1-Score of 0.103 across
all T cell subtypes (compared to only 0.045 across all cell types) (Supp. Fig. 8). In
summary, this example highlights the potential of our scTab model to distinguish between
closely related cell types and hints that the improvement of non-linear models will further
increase once more finely annotated training data becomes available.

**2. Following from the previous point, the authors explain how they have considered**
**ontology during the preparation of the training set. The level of ontology hierarchy or**
**granularity of cell type annotation is known to affect the performance of cell type classifiers**
**(see as an example the two levels of hierarchy in CellTypist “Immune_All” models).**

**Author response:**

We thank the reviewer for this question. We grouped our answer into two parts. First, we provide
more details about how we considered the Cell Ontology during the preparation of the training
data. Second, we investigated how classification performance is impacted by using more coarse
cell type labels.

For our benchmarks, we used the most fine-grained annotation given by the author associated with
the CL term as specified by the CELLxGENE schema¹. Note that upon submission of datasets to
CELLxGENE, authors must associate their cell annotations with the “closest CL term” which exists
at the time of submission. When preparing the training data we removed too coarsely annotated
cell types from our training data, i.e. those CL terms that had less than seven parent nodes in the
cell type ontology graph, as those cell types correspond to very coarse cell type labels that do not
contain much information.

Next, to test how much ontology level selection impacts scTab’s performance, we evaluated the
performance of our scTab model on a set of more coarsely annotated cell type labels. We selected
coarse cell type labels based on the information content score provided by the cell ontology
(<https://github.com/INCATools/ubergraph?tab=readme-ov-file#graph-organization>). The information
content score is calculated based on the count of terms related to a given cell ontology term and is
in the interval [0, 100], where 100 corresponds to a very specific term with no subclasses. Based
on the information content score we used the following rules to define a set of coarse cell type
labels:

- 1. Get all cell type labels present in the CELLxGENE census which are a subset of the “native
cell” cell type label
- 2. Keep all cell type labels with an information content score of less or equal to 60

- 3. Assign each cell type label to one of the coarse cell type labels from step 2. If based on the
 cell type ontology, a cell type label can be assigned to more than one of the coarse labels,
 we only assign it to the coarse label with the highest information content score. Example:
 the label “alpha-beta T cell” would be assigned to “T cell” as the coarse label and not
 “lymphocyte”.
- 4. Use the grouping from Step 3 to assign each of the fine-grained cell type labels to a coarse
 cell type label.

 The resulting coarse cell type labels and the corresponding classification performance (measured
 by F1-Score) can be found in the plot below (or Supp. Fig. 13):

 For the coarse cell type labels defined above, scTab achieves a macro F1-Score of 0.897
 compared to 0.830 on the fine-grained annotations. This trend is not surprising, as we expect
 classification between coarse categories of cell types (with distinct transcriptional signatures)
 should be easier than classification between refined subtypes of cell types (which share similar
 molecular signatures). This is also in line with the observations from Major Comment #1 by
 Reviewer #1 whereby one can see that distinguishing T cell subtypes is a much more difficult
 problem than just identifying T cells in general. Lastly, we would like to note that we did not retrain
 the model from scratch for the evaluation on the coarse cell type labels. Instead, we aggregated
 the predictions of the model that was trained on the fine-grained cell type labels. For instance, all
 predictions of “mature T cell” subtypes count as predicting the label “mature T cell” (based on the
 underlying hierarchy of the Cell Ontology). Note that this also explains the poor performance for
 some of the coarse cell types e.g. “hematopoietic stem cell”, “squamous epithelial cell” or “blood
 cell” where either only a single or just two fine-grained cell type labels get grouped together to form
 a coarse cell type label. The results, in this case, are basically the same evaluation as if one would
 use the fine-grained cell type labels.

 We’ve added more clarifying details to the manuscript:

Second, we removed cells with broad cell type labels, using a heuristic that removes each
cell type with less than seven parent nodes in the Cell Ontology. For the remaining cell
types, we used the most fine-grained labels given by the author associated with the CL
term as specified by the CELLxGENE schema and did not map the author-provided cell
types to more coarse Cell Ontology terms.

and, additionally, added the performance evaluation on the coarse cell type labels:

Related to the previous point, we asked how our scTab model would perform on more
coarse cell type labels. To do so, we mapped the 164 original cell type labels to 31 coarse
cell type labels (Methods). Evaluated on the 31 coarse cell type labels, scTab achieves a
macro F1-Score of 0.897 compared to 0.830 on the fine-grained annotations. This trend is
not surprising, as we expect classification between coarse categories of cell types (with
distinct transcriptional signatures) should be easier than classification between refined
subtypes of cell types (which share similar molecular signatures).

**3. In lines 222-224, the authors state: “Furthermore, one can see that classes on which**
**errors were made tended to be those that were represented by few observations (Fig. 1g).”**
**Interestingly, the scatterplot in Fig. 1g also shows that a group of cell types with low cell**
**counts show F1-scores as high as more abundant cell type classes. Can the authors**
**comment on their interpretation of this phenomenon and whether this affects specific cell**
**lineages or higher levels of cell type ontology terms?**

**Author response:**
We wish to thank the reviewers for this question, as this trend deserves more exploration in the
original manuscript.

We include the scatter plot in Figure 1-G here for ease of review:

To explore this phenomenon of “easy-to-classify” cell types, i.e. cell types with low sample sizes in
the training data yet relatively high F1-scores, we list in the table below cell types with an F1-Score
of above 0.9 and with less than 10000 cells in the training data. (We’ve chosen 10000 cells as

having a small sample size, in line with the reviewer’s question). The values are sorted by
 F1-Scores in descending order:

	F1-score	Number of cells
chandelier pvalb GABAergic cortical interneuron	0.993885	7268
L6b glutamatergic cortical neuron	0.9928	9727
caudal ganglionic eminence derived GABAergic cortical interneuron	0.973098	6299
retina horizontal cell	0.971226	8530
cardiac neuron	0.967201	4629
lung pericyte	0.959292	3254
bronchus fibroblast of lung	0.956621	4299
ependymal cell	0.948148	4709
paneth cell	0.946496	3328
Bergmann glial cell	0.924838	4564
inflammatory macrophage	0.923077	8629
alternatively activated macrophage	0.917476	7041
renal interstitial pericyte	0.902137	4455

 Based on the identity of the cell types alone, we weren’t able to discern any clear trend. It is true
 that neuron subtypes appear more frequently but this covers only about half of the cell types. The
 overall fraction of neuron subtypes in the table above is around 39%, whereas the fraction of
 neuron subtypes across all of the training data is around 11%. This shows that neuron subtypes
 appear more frequently in the table above compared to what would be expected based on their
 overall frequency in our dataset. However, given the sizable presence of brain datasets in our
 training data, a special trend here doesn’t appear particularly convincing.

 Moreover, as a reference group, we list in the table below cell types with low F1-scores and low
 sample sizes, i.e. cell types with an F1-Score of below 0.4 and with less than 10000 cells in the
 training data. This serves as a reference group as those cell types have a similar amount of
 training data yet a significantly lower F1-Score. The values again are sorted by F1-Scores in
 descending order:

	F1-Score	Number of cells
double negative T regulatory cell	0.393056	3843
erythroblast	0.351064	5593
innate lymphoid cell	0.332346	6214
intermediate monocyte	0.332178	4082
megakaryocyte-erythroid progenitor cell	0.328518	5346
respiratory hillock cell	0.307177	4270

 These are indeed blood cells, but nearly half of our training data consisted of blood cells.

 As there isn’t any clear trend visible as to why some cell types can be classified well based on very

little training data whereas others can not, we hypothesized whether a more general explanation
might be that the easy-to-classify cell types have a specific gene expression signature. That is, the
distinct transcriptomic signature of these cells would make these cells easier to classify and equally
important generalize well from the training data to the test data. In contrast, the hard-to-classify cell
types have a gene expression signature that either makes them hard to distinguish from other
similar cell types or that does not generalize well from the training to the test data.

To investigate this further, we would first like to highlight how the TabNet architecture can be used
to understand which features are used to classify specific cell types. The TabNet architecture uses
learned feature attention masks - those masks are sample-specific and are usually very sparse
(only about 1% of the mask entries are non-zero). By looking at the feature attention masks, one
can see which genes are used as features to classify certain cell types. Consequently, for our
analysis, we used the top 200 genes (based on the feature attention masks) for each cell type and
investigated how well each cell type listed in the tables above can be separated from all other cell
types in the training data based on those top 200 “marker genes”. In the figure below (Supp. Fig.
9), one can see that the easy-to-predict cell types (marked in blue in the plot below) can be
classified pretty well based on the 200 “marker genes” alone (measured by the area under the
curve under the precision-recall curve), suggesting that those cell types have a pretty specific gene
signature, which gets picked up by the model. But more importantly, this gene signature
generalizes well from the training to the test data: The classification performance only drops
minorly from the training to the test set. On the other hand, for the hard-to-predict cell types
(marked in orange in the plot below), the classification performance on the training set is already
quite a bit worse. This suggests that those cell types are already harder to distinguish from the
other cell types in the training set. Additionally, when comparing the average drop in F1-Score
between the easy and hard-to-classify cell types, one can see that the genes selected by the
model for the hard-to-classify cell types generalize a lot worse to the test set compared to the
easy-to-classify cell types. For the easy-to-classify cell types the average F1-Score drops only
around 0.03 whereas for the hard-to-classify cell types the average F1-Score drops around 0.32.

In summary, one can see that the easy-to-predict cell types seem to have a unique gene
expression profile that on the one hand can be described by a few genes (200 genes for our

experiments) and that more importantly generalize well from the training to the test set. Hence,
making those cell types easy to classify even if only limited training data is available. This
phenomenon seems to be enriched for neuron subtypes in our training data but is not limited to
neuron subtypes.

Lastly, we want to look into the genes we selected based on the TabNet attention masks in a bit
more detail. To calculate the attention-based ranking of genes for each cell type we first calculate
the sample/cell-specific attention masks, then multiply the attention scores with the corresponding
input gene matrix. This corresponds to the input the model uses to classify each cell/sample. In the
last step, we aggregate those sample/cell-specific attention scores by taking the mean across all
the cells for a specific cell type. This gives a feature importance score for each gene grouped by
cell type. One can see the resulting top 25 “marker genes” (sorted by attention score in descending
order) for the “easy-to-classify” cell types in Supp. Table 10.

We wanted to check that these genes were indeed relevant to the biological predictions of the cell
types. We did indeed find biologically relevant genes in the predictions:

**Macrophages:**

- ● *Detailed list of cell types:* “alternatively activated macrophage”, “alveolar macrophage”,
“elicited macrophage”, “inflammatory macrophage”, “lung macrophage”, “macrophage”
- ● Genes in attention masks (which appear across all macrophages):
 - ○ **B2M**
 - ■ *Biological significance:* Related to the immune system, MHC class I
 - ■ *Reference:* <https://www.genecards.org/cgi-bin/carddisp.pl?gene=B2M>
 - ○ **FCGR3A**
 - ■ *Biological significance:* The immune system: This gene encodes a receptor
for the Fc portion of immunoglobulin G
 - ■ *Reference:* <https://www.genecards.org/cgi-bin/carddisp.pl?gene=FCGR3A>
 - ○ **HLA-DRA** and **HLA-DQB1** only for “inflammatory macrophage”
 - ■ *Biological significance:* The immune system, HLA
 - ■ *Reference:* <https://www.genecards.org/cgi-bin/carddisp.pl?gene=HLA-DRA>

**Central nervous system related cells:**

- ● Genes in attention masks:
 - ○ **NRXN1**
 - ■ *Detailed list of cell types:* “L6b glutamatergic cortical neuron”, “caudal
ganglionic eminence derived GABAergic cortical interneuron”, “chandelier
pvalb GABAergic cortical interneuron”, “central nervous system
macrophage”
 - ■ *Biological significance:* Associated with the Central Nervous System: This
gene encodes a single-pass type I membrane protein that belongs to the
neurexin family. Neurexins are cell-surface receptors that bind neuroligins to
form Ca(2+)-dependent neurexin/neuroligin complexes at synapses in the
central nervous system
 - ■ *Reference:* <https://www.genecards.org/cgi-bin/carddisp.pl?gene=NRXN1>
 - ○ **NTM** only for “cardiac neuron”

- ■ *Biological significance:* Related to Jacobsen syndrome (heart defects), and
- neurite outgrowth
- ■ *Reference:* <https://www.genecards.org/cgi-bin/carddisp.pl?gene=NTM>
- ○ **SYT1**
- ■ *Detailed list of cell types:* “Bergmann glial cell”, “L6b glutamatergic cortical
- neuron”, “caudal ganglionic eminence derived GABAergic cortical
- interneuron”, “chandelier pvalb GABAergic cortical interneuron”, “ependymal
- cell”, “central nervous system macrophage”, “retina horizontal cell”
- ○ *Biological significance:* Calcium sensor that participates in triggering
- neurotransmitter release at the synapse
- ○ *Reference:* <https://www.genecards.org/cgi-bin/carddisp.pl?gene=SYT1>

**Retina horizontal cell:**

- ● Genes in attention masks:
- ○ **FRY**
- ■ *Biological significance:* Plays a key role in patterning sensory neuron
- dendritic fields
- ■ *Reference:* <https://www.genecards.org/cgi-bin/carddisp.pl?gene=FRYL>
- ○ **CACNA1A**
- ■ *Biological significance:* Neurotransmission
- ■ *Reference:* <https://www.genecards.org/cgi-bin/carddisp.pl?gene=CACNA1A>
- ○ **KIF1B**
- ■ *Biological significance:* Chemical synaptic transmission
- ■ *Reference:* <https://www.genecards.org/cgi-bin/carddisp.pl?gene=KIF1B>

We've added the changes below to the manuscript discussing this phenomenon:

**Another important observation is that** classes on which errors were made tended to be

those that were represented by few observations (Fig. 1g). This further motivates that

classification performance can indeed be improved by adding more training examples

specifically for cell types that are hard to classify. **When looking at the plot in more detail,**

**one can see that some cell types achieve quite remarkable classification performance**

**despite very little available training data. To investigate this further, we looked at the cell**

**types that have an F1-score of 0.9 or higher and that have less than 10,000 cells in our**

**data corpus (Supp. Table 9). Looking at the table, one can see that this phenomenon**

**appears to be enriched among cell types associated with the central nervous system.**

**However, it's not limited to those. Thus, we hypothesize that there is a more general**

**explanation for this. Namely, those cell types have a specific gene signature that on the one**

**hand makes the cell types easy to distinguish from other cell types and on the other hand**

**generalizes well from the training to the test data. To investigate this hypothesis in more**

**detail, we look at the feature attention masks of the scTab model to determine which**

**features are important when generating predictions. The feature attention masks are sparse**

**(they are non-zero for only around 1% of the gene space) and indicate which genes the**

**model prioritizes when classifying a specific cell or cell type. We selected the top 200 genes**

**(~1% of the total gene space) ranked by the feature attention scores of the scTab model**

**and looked at how well a linear model can separate a cell type from all other cell types**

**purely based on those top 200 genes. We then compared the separation scores (measured**

by the area under the precision-recall-curve) to a reference group that included all cell
 types with an F1-score of less than 0.4 and again less than 10,000 cells of available
 training data (Supp. Fig. 9). One can see that the “easy-to-predict” cell types are easier to
 distinguish based on the selected top 200 genes and that, more importantly, the gene
 signature generalizes well from the training to the test data (for the “easy-to-predict” cell
 types the F1-scores only drop around 0.03 versus 0.32 for the reference group). Moreover,
 we checked whether the genes in the feature attention masks do indeed correspond to the
 biology of the predicted cell type. We found that these gene signatures appear not to be
 random. The top 25 genes as ranked by feature attention scores were often found to be
 biologically relevant: as a brief example, the top genes for predicted brain cells include
 genes encoding neurexin and synaptotagmin proteins (e.g. *NRXN1*, *SYT1*), the top genes
 in the feature attention masks for all predicted macrophages included genes related to the
 immune system (e.g. *FCGR3A*, *HLA-DRA*), and so forth.

 **4. Could the authors show a runtime comparison of the training with the models compared**
 **in Fig. 1c?**

 **Author response:**

We thank the reviewer for the valuable suggestion. We added a runtime comparison in the table
 below (which can be found in the newly added Supp. Table 8). Additionally, we added a
 comparison of the inference speeds of the different models. Inference speed is probably the more
 interesting metric for models of our size, as the trained models are mostly used for inference and
 training such large-scale models will always require specialized hardware and will be quite
 resource-intensive.

	Training time [in hours]	Inference time [samples/sec]
CellTypist (trained on 1.5 Mio cells)	~16h (AMD EPYC 7642 - 20 cores)	~2000 (AMD EPYC 7642 - 20 cores)
Optimized linear	~20h (1x Nvidia A100 40GB)	~29500 (1x Nvidia V100 16GB)
scTab	~33h (1x Nvidia A100 40GB)	~10800 (1x Nvidia V100 16GB)
XGBoost	~10h (1x Nvidia A100 80GB)	~4200 (1x Nvidia V100 16GB)
MLP	~29h (1x Nvidia A100 40GB)	~21400 (1x Nvidia V100 16GB)

 Explanation of different hardware used:

- ● CellTypist only supports CPUs. Hence, we could not use any GPU to speed up training or
 inference.
- ● XGBoost has very high GPU memory requirements as it is not trained in batches, unlike the
 other methods. Therefore, we had to use A100 80GB to be able to train the model
- ● Inference benchmarks are done on V100 GPUs as more compute wasn't needed for
 inference and they are more available on our HPC cluster.

 **5. In Fig 2e, the authors show the comparison in performance in cell classification of**
 **organ-specific training and cross-tissue training with lung data, indicating that**
 **”cross-organ cell type classification benefits from using non-linear models”.** To make this

statement, could this comparison be shown for all individual tissues or, at least, a higher
number of per-tissue models?

**Author response:**

We agree with the reviewer; indeed, this should be explicitly shown on a wider selection of tissues,
and detailed for the reader.

We begin by plotting the F1-Score performance into 14 tissue categories as organized by major
research efforts within the Human Cell Atlas, the Biological Networks:
<https://www.humancellatlas.org/biological-networks/>

To illustrate the strength of using cross-organ data for training, we selected the tissues where there
is a strong difference in F1-Score performance between scTab and the optimized linear model (see
barplot below).

Based upon these results, we selected the following tissue categories for downstream analysis:
lung, kidney, heart, gut, breast, blood + immune.

In the plots of F1-Score performance below, we demonstrate how organ-specific performance
impacts scTab and the optimized linear model trained on either only organ-specific data or on the
aggregate data of all organs combined (i.e. the cross-organ setting). One can see that the
non-linear scTab model is more robust when going from organ-specific to cross-organ models i.e.
classification performance is less affected compared to the optimized linear reference model. This
highlights the value of non-linear models when training cell type classifiers across a wide range of
organs.

We've added these results to the main manuscript:

**6. In connection with the previous point, could the authors show whether the improvement**
 **between cross-organ and individual tissue models is driven by specific cell type classes?**
 **Do the classes driving the improvement in performance change according to tissue? Are**
 **misclassified cells always the same cell type? The authors could provide a scatter plot**
 **showing the correlation between per-class F1-scores in scTab versus the optimized linear**
 **model and show the actual cell type labels that vary the most between models.**

**Author response:**

We thank the reviewer for this suggestion. This is useful to clarify.

model versus the ones from the optimized linear model:

 Given limited space in the figure and the often long cell type labels, for readability we have
 reported the top 7 cell types with the biggest improvement over the optimized linear model
 (highlighted in green) and the top 3 cell types with the worst classification performance of the
 scTab model (highlighted in red) in the table below:

	Cell types with the biggest difference in F1-Score (scTab vs. Optimized linear)
Lung	 ● CD4-positive helper T cell ● T-helper 17 cell ● Schwann cell ● CD16-positive, CD56-dim natural killer cell, human ● CD8-positive, alpha-beta cytotoxic T cell ● vascular associated smooth muscle cell ● effector memory CD8-positive, alpha-beta T cell ● intermediate monocyte ● naive thymus-derived CD8-positive, alpha-beta T cell ● effector CD8-positive, alpha-beta T cell
Kidney	 ● pericyte ● conventional dendritic cell ● alternatively activated macrophage ● monocyte ● CD4-positive, alpha-beta T cell ● non-classical monocyte ● renal interstitial pericyte ● CD4-positive helper T cell ● CD8-positive, alpha-beta T cell ● capillary endothelial cell
Heart	 ● lymphoid lineage restricted progenitor cell ● Schwann cell

	 ● mature NK T cell ● vein endothelial cell ● immature innate lymphoid cell ● capillary endothelial cell ● Lymphocyte ● vein endothelial cell ● B cell ● endothelial cell of artery
Gut	 ● effector memory CD8-positive, alpha-beta T cell, terminally differentiated ● erythroblast ● myoepithelial cell of mammary gland ● T follicular helper cell ● CD16-negative, CD56-bright natural killer cell, human ● CD8-alpha-alpha-positive, alpha-beta intraepithelial T cell ● precursor B cell ● alveolar macrophage ● inflammatory macrophage ● T-helper 17 cell
Breast	 ● myoepithelial cell of mammary gland ● gamma-delta T cell ● naive B cell ● effector memory CD8-positive, alpha-beta T cell ● mature NK T cell ● activated CD8-positive, alpha-beta T cell ● class switched memory B cell ● CD8-positive, alpha-beta T cell ● B cell ● memory B cell
Blood and immune	 ● pro-B cell ● immature innate lymphoid cell ● precursor B cell ● promonocyte ● effector memory CD8-positive, alpha-beta T cell, terminally differentiated ● IgA plasma cell ● activated CD4-positive, alpha-beta T cell ● lymphoid lineage restricted progenitor cell ● capillary endothelial cell ● pericyte

Looking at the table above, one can see that again as in Major Comment #1 by Reviewer #1, “T

cell” subtypes appear to be enriched when looking at the cell types with the biggest improvement

over the linear model (except for the kidney). However, the specific T cell subtypes usually change

with the different tissues. Furthermore, there isn’t any clear trend visible in the cell types on which

our scTab model performs worst.

As the results above are similar to those addressed in Major Comment #1 by Reviewer #1 we did

not make changes to the manuscript. Moreover, we added some clarifying remarks regarding

Figure 2e.

**Note:** In passing, we wanted to point out that we noticed above the cell annotation “myoepithelial

cell of mammary gland” cell type in the “gut” is incorrect, as it is a breast-specific cell type. This

looks like an annotation error by the authors from the underlying CELLxGENE dataset

(<https://cellxgene.cziscience.com/e/4ed927e9-c099-49af-b8ce-a2652d069333.cxg/>) where the
authors assigned this cell type both in the “breast” as well as the “esophagus mucosa”. The same
applies to the “alveolar macrophage” cell type in the “gut”, which appears in the “colon”, “omentum”
and “spleen” as well. However, unlike for the previous dataset, this only affects very few cells.
(<https://cellxgene.cziscience.com/e/1b9d8702-5af8-4142-85ed-020eb06ec4f6.cxg/>).

This is another example whereby the training data does contain errors, which will be revised in the
future with other HCA initiatives like the Cell Annotation Platform.

**7. A more detailed explanation of common use cases for the scTab pre-trained models in**
**different settings would be advisable given the broad interest in the single-cell community.**
**Also, an indication of use cases for newly trained models.**

**Author response:**

We thank the reviewer, as this is indeed something we could clarify.

These pre-trained model checkpoints and the associated Python code are meant to be accessible
by computational biologists who wish to annotate their datasets. We’ve created brief tutorials on
how to perform these tasks in the GitHub repo, e.g.
https://github.com/theislab/scTab/blob/devel/notebooks-tutorials/model_inference.md

As we attempt to motivate in the manuscript’s introduction, manually annotating datasets is a fairly
time-consuming and error-prone task. Providing these model checkpoints and Jupyter notebooks
for the community should allow researchers to quickly annotate their datasets, with the F1-score
used as a confidence metric. We encourage biologists to use these predictions to further refine and
correct the predicted cell annotations using standard single-cell analysis software.

We also have provided reproducible Python code for users to train models on other large-scale
collections of scRNAseq data which may be collected in the future. Although we used the
CELLxGENE Census as training data, this does not need to be human-specific; we suspect in the
coming years large collections of datasets of model organisms will be collected similarly.

Moreover, to be easily accessible to all biologists with a web browser, there is currently
development to integrate scTab into the Cell Annotation Platform, such that users can annotate
datasets online with these automated predictions via a UI. With these cell annotations stored in a
standardized way, this will provide more refinements for future model predictions. Every few
488 months, given the more cell annotations ingested, the model’s predictions could be refined for
users to work with.

We have added more clarifying details to the manuscript.

**8. The authors made available both the training data as well as the code for generating the**
**scTab models at <https://github.com/theislab/scTab>. It is however unclear if the authors**
**intend their pre-trained models to be used by the community. If so, instructions on how to**

use the scTab models for prediction on new datasets should ideally be included as part of
the documentation.

**Author response:**

We thank the reviewer for the great suggestion, we added a tutorial to our GitHub repository and
added a reference to the tutorial in the “Code and data availability” section:

https://github.com/theislabs/scTab/blob/devel/notebooks-tutorials/model_inference.ipynb

The notebook is grouped into three parts:

1. Preprocessing of the input data:

a. Streamlining/aligning the feature space. This means selecting only the genes the
model was trained on and ordering the genes in the same way as the training data.

b. Wrapping the resulting anndata.AnnData object into a PyTorch dataloader to be
able to use the data with our model.

2. Loading the pretrained model weights and initializing the model for inference.

3. Running model inference:

a. Running inference with the pretrained model

b. Mapping the model predictions, which are integers, to their corresponding cell type
label.

Naturally, the reviewer’s comments here relate to the previous question regarding use cases: we
are integrating this model into the Cell Annotation Platform for users to interactively annotate their
datasets with predictions as hints.

**9. Only 10X Genomics data is used for training the models. Could the authors comment on**
**the rationale for this and could they provide insights into performance across chemistries**
**given that CELLxGENE already contains non-10X Genomics datasets?**

**Author response:**

Thank you for this question. Reviewer #3 brings up the same point.

Indeed, this is an interesting question. To answer this, we grouped our response into two parts. In
the first part, we added more explanation on our decision why we limited the training and
evaluation data only to data from 10X-based sequencing protocols. In the second part, we
evaluated our model, that was trained only 10X-based data, on the non-10X-based data from
CELLxGENE to give the reader an idea how our model would perform in this setting.

We decided to limited our training data only to data from 10X-based sequencing protocols as the
amount of non-10X-based data on CELLxGENE is currently very limited. We visualized this in the
537 bar plot below where the number of unique donors is plotted for each sequencing protocol:

 The data from 10X-based sequencing protocols is clearly far more prominent compared to all other
 non-10X-based protocols. One of the key objectives of our paper is to train a cell type classifier
 across a wide range of donors/datasets. This is simply not currently available to us for
 non-10X-based protocols, as we've only been able to find sequencing from a couple of dozen
 unique donors. Given this strong imbalance, we decided to limit the training and evaluation data
 only to 10X-based data as there is simply not enough data available to reliably train and evaluate
 our model on non-10X-based data. Once non-10X-based data becomes available at a similar
 scale, it would indeed make sense to include it.

 In passing, we note that it appears the single-cell community is simply not sequencing non-10x
 protocols at the same magnitude, or at least this isn't publicly available to us as training data. The
 CELLxGENE data corpus is clearly focused on 10X-based data. Furthermore, when comparing the
 increase in number of donors of the CELLxGENE census release we used in our paper
 ("2023-05-15") to the latest LTS (long-term support) census release ("2023-12-15"), one can again
 see a focus on 10X-based data: for 10X-based data, there were an additional 378 donors added,
 whereas the non-10X-based data even decreased by 48 donors. Therefore, we just don't have the
 data based on non-10x-based protocols to do this analysis properly, and as a consequence, we
 didn't focus on this topic in the manuscript.

 Nonetheless, to explore how our model would perform on non-10X-based data, given our current
 training data which only includes 10X-based data, we evaluated our current model on the
 non-10X-based data from CELLxGENE, subsetting only the cell types that are present in the
 training data to do a more accurate comparison. As a first step, we calculated the macro F1-score
 grouped by sequencing protocol to give an impression of how well our model generalizes to
 different sequencing protocols; see the bar plot below:

 One can see that we get decent classification performance on about half of the sequencing
 protocols with a macro F1-Score of ~ 0.4 . In comparison, the macro F1-Score on the holdout
 10X-data is ~ 0.8 . This is quite impressive, given that in this case the model has to generalize to
 unseen datasets which are measured with a different sequencing protocol - meaning there is a
 much stronger shift in data distribution compared to the donor holdout evaluation setting we used
 in our paper. However, there are also sequencing protocols to which our model generalizes quite
 poorly (namely “STRT-seq”, “Smart-seq2” and “BD Rhapsody Targeted mRNA”), indicating an even
 stronger shift in data distribution for those sequencing protocols.

 In the second step, we look at per-cell-type classification performance. As the classification
 performance can vary strongly depending on the sequencing protocol, we calculated the
 classification performance per sequencing protocol separately and then reported the mean and
 95% confidence intervals for the F1-Score of each cell type in the barplot below. Looking at the plot
 one can see that the model trained only on 10X-based data can achieve descent classification
 performance for some sequencing protocols. However, the performance varies strongly across the
 different sequencing protocols (note the very large confidence intervals), highlighting our initial
 point that more data is necessary to reliably train and evaluate our model on non-10X-based data
 as there are often only a few dozen donors available for many of the non-10X-based protocols.

We have added the section below to our manuscript to indicate to the reader how scTab would
perform on non-10X-based data and to provide more details as to why we limited the data to only
10X-based data:

Lastly, it is interesting to investigate how scTab, which is only trained on 10X-based training
data, would perform on data from non-10X-based sequencing protocols. Hence, we looked
at the classification performance (measured by macro F1-score) grouped by sequencing
protocol (Supp. Fig. 10). One can see that scTab achieves decent classification
performance on about half of the non-10X-based sequencing protocols with a macro
F1-Score of ~0.4. In comparison, the macro F1-Score on the holdout 10X-data is ~0.8. This
is quite impressive, given that the model has to generalize to unseen datasets that are
measured with a different sequencing protocol - meaning there is a much stronger shift in
data distribution compared to the donor-based holdout evaluation setting. However, there
are also sequencing protocols to which our model generalizes quite poorly (namely
STRT-seq, Smart-seq2, and BD Rhapsody Targeted mRNA), indicating an even stronger
shift in data distribution for those sequencing protocols. The above observations further
justify our decision to limit the training data only to 10X-based sequencing protocols. Given
the strong shift in data distribution between the different sequencing protocols and that the
data from CELLxGENE is clearly dominated by 10X-based sequencing protocols (Supp.
Fig. 11), it would make it challenging to train and evaluate our model on non-10X-based
data reliably.

**10. Universal Cell Embeddings (UCE) has been recently presented as a universal**
**representation of all cells in the human body that can be used for cell type annotation (DOI:**
**<https://doi.org/10.1101/2023.11.28.568918>).** **Could the authors comment on how this**
**publication relates to their work and whether it would be appropriate to include UCE in their**
**benchmarking?**

**Author response:**

Thank you to the reviewer for this question.

Despite being trained on a similar amount of data (36 million cells for UCE and 22 million cells for
our scTab model), the two models are fundamentally different. The most notable difference is the
amount of computing resources required. Our model is fitted for around one day on a single Nvidia
A100 GPU. In contrast, the UCE model is fit for 40 days on 24 A100 GPUs, hence using almost
three orders of magnitude more computing power - this made it impossible to fit with the computing
resources that are available to us, and unfortunately, we wouldn't be able to include this in
benchmarking. Furthermore, already running inference on the model is very costly. As mentioned
in question 8 from reviewer 3, running inference on the scGPT model took several hours. Given
that UCE has 650 million parameters vs. 53 million parameters for the scGPT model
(<https://github.com/bowang-lab/scGPT/issues/125>), it would already take several days just to get
the embeddings for the cells in our test data. Moreover, conceptually, the UCE model tries to learn
a universal cell representation independently from/without the use of cell annotations; thus the
output of the model is a 1280-dimensional embedding vector which can then be used to search for
similar cells in an annotated reference atlas by searching for the closest embedding vectors from
the reference atlas. This is a fundamentally different use case from our scTab model, which directly
predicts a cell type label based on the gene expression profile of a cell - making it independent

from a reference dataset/atlas, and distinct from annotation transfer approaches like UCE. Lastly,
as UCE focuses on learning cell embeddings, the evaluation of the cell type annotation
performance is often more qualitative and lacks more quantitative metrics, for example, measuring
the classification performance with the macro F1-Score. This would also make benchmarking for
our case quite tricky.

Nevertheless, to give the reader an idea of the performance of the UCE model, we used the UCE
embeddings which are hosted by CELLxGENE (<https://cellxgene.cziscience.com/census-models>)
and evaluated them in the linear probing setting. This means fitting a logistic regression model
based on the embeddings obtained by UCE. Unfortunately, these pre-computed embeddings only
exist for census version “2023-12-15” which is missing some datasets that were included in census
version “2023-05-15” (the census version we used for our paper). In numbers, this means, we
could only evaluate the UCE model on 736 of the 758 donors from our test data.

We fitted the logistic regression classifier on a random subsample of 1,500,000 cells of the training
data and then evaluated this classifier on the reduced test data (only 736 of the original 758 donors
from our test data). In this setting, UCE achieved a macro F1-score of 0.7611 ± 0.0018 .

For easier comparison, we plotted the classification performance (measured by macro
F1-score) of scTab, scGPT (zero-shot and fine-tuned), UCE, and 256-dimensional PCA
embeddings in the barplot below (or Supp. Fig. 6):

Finally, we would like to highlight again that it’s not possible for us to fine-tune a UCE model
due to its large computing requirements. And, as there are no tutorial notebooks or general
API available that explain how to fine-tune the UCE model to a specific downstream task.
Moreover, as the UCE model is trained on all data available on CELLxGENE, it has already
seen our holdout test data during the self-supervised pre-training. This potentially results in a
too-optimistic classification performance as our holdout test data isn’t true holdout test data for

the UCE model anymore (it has already seen those data points during the self-supervised
pretraining).

We have added the Universal Cell Embedding model to our benchmark in the paper in Supp.
Fig. 6.

Minor points:

**1. We would recommend an alternative representation for Fig1a. Most of the cell labels are
not readable and the relative contribution of cell types and tissues is not clear.**

We thank the reviewer for pointing this out. However, this is quite a challenging task as there are
164 cell type labels and over 60 tissue labels - making it almost impossible to summarize all the
information in the space available in the figure. So, instead, we wanted to show a broad overview
of the data in Fig. 1 a). To give a more detailed overview, we added summary bar plots for the
number of donors per cell type and the number of donors per tissue in the newly added Supp. Fig.
1 and reference to those summary plots in the figure caption.

**2. We would suggest modifying “Linear (ours)” by for example “Optimized linear”**

**Author response:**

We thank the reviewer, and we have modified the manuscript accordingly.

**3. We would suggest rewriting the following sentence in lines 89-91 as it is not
understandable: “However, those efforts on foundation models benchmark cell
representations on diverse tasks, so far without context on deep models specifically
designed for cell type annotation.”**

**Author response:**

We thank the reviewer and have tried revising the sentence to be clearer:

However, those efforts on foundation models mainly focus on learning cell
representations/embeddings in an unsupervised manner, without a specific focus on cell
type annotation (and especially without making use of author-provided cell type labels).

**4. In lines 98-100, the “zero-shot” concept appears for the first time, and it is mentioned
several times throughout the manuscript. We suggest including a brief definition of the
term: “Furthermore, according to recent benchmarks 25,26, recent foundation models often
only show comparable performance to simpler and often linear reference models in the zero
or few-shot setting.”**

**Author response:**

We thank the reviewer for pointing this out.

With the term “zero-shot”, we refer to what is described as “linear evaluation” in computer vision or
natural language processing. We adapted the term “zero-shot” as it is frequently used in the
context of single-cell foundation models like for example scGPT. The “zero-shot setting” means
directly using the embeddings of a pre-trained foundation model as input/features to a linear
classifier (e.g. logistic regression) for the downstream task we want to solve (In our case, this is
cell type classification). This setting is very computationally efficient, as it does not require any
model retraining; rather, it only requires training a simple linear model. The few-shot setting refers
to fine-tuning a pre-trained foundation model with only limited training data, in natural language
processing this means only using a few examples (hence the name), and in single-cell genomics,
this usually means a few thousand up to a few hundred thousand cells. This setting is much more
computationally expensive than the zero-shot setting as it requires fine-tuning/training the
underlying foundation model.

We have added a clarification to the manuscript.

**5. We suggest omitting the word “significant” in the following sentences:**

**Lines 196-199: “We implemented a logistic regression-based model not subject to these**
**limitations (Methods) and trained this model with a cross-entropy loss. This model**
**significantly outperformed the CellTypist reference model and achieved a macro F1-score of**
**0.7848±0.0001 (Fig. 1c)”.**

**Lines 204-206: “The nonlinear models significantly outperformed the linear model**
**(0.8295±0.0007 macro F1-score for scTab (fitted with data augmentation), 0.8127±0.0005 for**
**XGBoost, 0.7971±0.0012 for MLP (fitted with data augmentation)) (Fig. 1c, Supp. Table 1)”.**

**Author response:**

We agree with the reviewer and we have removed the term “significantly” in these cases.

**6. If Fig. 2b, neither the upper title: “number of unique donors” nor the lower title: “number**
**of training samples”, seems to indicate the number of cells sampled. Did the authors intend**
**to indicate the “number of cells in the training set” in the lower title?**

**Author response:**

We thank the reviewer for pointing this out. We’ve corrected the title.

**7. Lines 63-65: We would suggest omitting the word “surprisingly”:**

**“However, cross-tissue classifiers trained on large-scale data collections that annotate cells**
**from heterogeneous sources, irrespective of tissue of origin and assay type, are**
**surprisingly slow to emerge.”**

**Author response:**

We agree with the reviewer and we have removed the term “surprisingly”.

**8. Reference 19 has been cited as a preprint but is now published in Nature Methods.**

**Author response:**

Thank you, we have updated the citation.

**Reviewer #2**

I co-reviewed this manuscript with one of the reviewers who provided the listed reports as
part of the Nature Communications initiative to facilitate training in peer review and
appropriate recognition for co-reviewers.

**Author response:**

Thank you for your contribution to this round of reviews!

**Reviewer #3**

The authors have developed a deep-learning-based classification algorithm for annotating
cell-type information, leveraging large-scale reference datasets. This method, adapted from
the TabNet architecture, was originally designed for tabular data with adaptation made for
its application to scRNA-seq data. scTab integrates feature and attentional transformer
blocks for feature selection.

While the manuscript is generally well-written and engaging, I have some reservations
about the novelty, significance, and generalizability of the proposed methods. A more
in-depth discussion comparing scTab with existing methodologies in this field would be
needed to enhance the manuscript's contribution to the scientific community. Additionally,
the choice of evaluation metrics for scTab warrants further elaboration. The details of my
comments are below:

**Author response:**

Thank you for the summary and constructive comments, which we address below.

Major Comments:

**1. The methods presented in the manuscript extend from currently available methods**
**primarily designed for tabular data. I appreciate the authors' efforts in adapting these**
**methods to the scRNA-seq context, particularly in terms of handling large-scale datasets.**
**However, there are several key challenges that seem to remain unaddressed, which affects**
**the perceived novelty and interest in the methods. For example:**

**(Line 108) The author discusses the potential overlap between labels (e.g., 'naive B cell' and**
**'B cell' in Figure 1), a common issue in real datasets. The approach to address this involves**
**adopting CELLXGENE as a root dataset. However, it appears that scTab does not**
**fundamentally address the hierarchical structure of the labeling inherent to such datasets.**

**Author response:**

Thank you to the reviewer for bringing this up. It's an important issue.

As we discussed in the response to Reviewer #1, Major Comment 2, the training data we used was
curated by CELLXGENE. Part of the submission process to CELLXGENE is that authors need to
associate their cell annotations to the "closest Cell Ontology term" which exists at the point of
submission. Given this was the largest training data that existed at the time, we used this for the
model.

There are major assumptions regarding how the training data was curated: that the hierarchies in
the CL are correct and that the CL term chosen by a scientist is consistent for all researchers. Both
of these assumptions are incorrect. As we briefly note in the discussion section of the manuscript,
these problems are meant to be rectified with the HCA's Cell Annotation Platform. For this
manuscript however to introduce the model, we take these two assumptions to be true given that's
the training data at scale which existed.

Moreover, we would like to highlight that this is an issue inherent to the CELLxGENE training data,
there is simply information missing if a cell is just labeled as a “B cell” instead of e.g. a “naive B
cell”. Such an issue cannot be solved by a change in model architecture as this would always
mean filling/predicting missing information but needs further data curation efforts to collect the
missing information. This, however, would be far beyond the scope of this paper. However, despite
training our model with not perfect training data, our model achieves considerable classification
performance (macro F1-Score of 0.83 on our holdout test set). This highlights the point that the
issue of missing information is not as pressing as one might think as one can still train a useful
classifier. To give a bit more intuition for this, one can look at the learned features / latent space of
our scTab model (output of the last layer based on which the cell types are classified with a linear
layer) in the tSNE plot below:

One can see that similar/related cell types group together in the latent space of our scTab model
through the cross-entropy loss - meaning the hierarchical structure of cell types labels is preserved
and hence the not perfect training data should at most result in slightly off predictions e.g.
predicting a cell to be a “B cell” instead of a “naive B cell” but not in completely wrong predictions.

We have added the text below to the discussion section of the manuscript:

Related to the previous issue, is the difference in annotation granularity across datasets on
CELLxGENE, some authors might annotate a cell as a “B cell” in their respective dataset
whereas another author might annotate the cell as a “naive B cell”. This issue is an inherent
issue with the CELLxGENE training data, as there is simply information missing in that
case. Such an issue cannot be solved by a change in model architecture as this would
always mean filling/predicting missing information but needs further data curation efforts to
collect the missing information. This, however, would be far beyond the scope of this paper.
Nevertheless, despite the potential issues with the training data, our model achieves
considerable classification performance (macro F1-Score of 0.83 on our holdout test set).

This highlights the point that the issue of missing information is not as pressing as one
might think as one can still train a working/useful classifier.

**2. One of the main contributions of scTab is its generalizability, which is facilitated by the**
**heterogeneity of cells provided by the large-scale training corpus. However, I believe its**
**generalizability has not yet achieved its maximum potential, and several improvements are**
**needed.**

(Line 153) The authors limit the model to a single genomic sequencing platform (10X
Genomics). However, it's unclear how the method would perform with query data from
different or mixed platforms. Can the authors clarify if the method would still be effective
when the training and testing data originate from diverse sequencing platforms? It would be
beneficial to conduct experiments to assess how discrepancies in data platforms might
influence performance.

**Author response:**

Thank you for this point. Reviewer 1 raises the same point in comment 9).

We decided to limit the training data only to 10X-based data as the CELLxGENE data corpus
consists mainly of 10X-based datasets. Thus, it is difficult to reliably train and evaluate our model
on non-10X-based data. Nevertheless, we added an evaluation of our model, which was trained on
10X-based data, on the non-10X-based data from CELLxGENE to give the reader an impression of
how well our model would perform on non-10X-based data.

Please refer to comment 9) from reviewer 1 for a more detailed evaluation.

**3. The scTab is designed to focus on cross-tissue classification. However, I am curious**
**about how the method addresses batch effects or differences arising from each tissue,**
**beyond simply merging the datasets. Could the authors elaborate on any specific strategies**
**or techniques employed by scTab to mitigate batch effects inherent in cross-tissue**
**analyses?**

**Author response:**

One of the main ideas behind our paper is to investigate whether the core ideas behind modern
deep learning could be applied to single-cell genomics data - or more precisely cell type
classification. Modern deep learning builds upon the idea of learning a decision function purely
based on data. This idea brought stunning breakthroughs in the fields of computer vision and
natural language processing, where the approach of using big models and large-scale training
datasets to regularize those models drastically outperforms other traditional (usually
feature-engineering-based) machine learning approaches. Currently, models for cell type
classification tasks based on annotation transfer are usually trained on limited training data (often
only a few 100MB in size), and are prone to overfitting to specific datasets, with one exception
provided models with small model capacity or by incorporating task-specific domain knowledge
(feature-engineering, enforcing batch correction, etc.). But as more training becomes available,
one can relax those constraints and move more in the direction of techniques that have proven
extremely successful in the areas of modern computer vision or natural language processing

(NLP). However, we would argue that it does not make sense to directly scale to model sizes that
are currently state-of-the-art in computer vision or NLP, as there is significantly more data available
to train those models. Hence, instead, we focused on whether we could observe similar results as
could be seen in e.g. the AlexNet paper in computer vision which was one of the key milestone
papers that shifted the field of computer vision towards what is now considered state-of-the-art in
computer vision.

Another important point is to distinguish classification-based models from autoencoder-based
models, such as scVI. Autoencoder-based models reconstruct the full gene expression profile from
their learned latent representation; by design, these models, therefore, include batch effects into
the latent space unless explicitly correcting for them, as the reconstruction objective function tries
to reconstruct as much variation in the original data as possible (i.e. the original data - *including*
batch effects). In contrast, models trained for classification (such as scTab) do not reconstruct the
input features but learn to extract features useful to distinguish cell types, which by design means
removing batch effects and biological variation that is not cell type specific as the cross entropy
loss function trains the neural network to map cells with the same cell type close together in the
latent space. That is how scTab accounts for batch effects. To give a visual motivation, we added
Fig. 1f and Supp. Figure 4 to the manuscript. The two figures show that scTab does achieve a
more structured/batch-corrected latent space. However, we would like to emphasize that the above
should be seen as a sanity check and not as a claim that scTab perfectly removes batch effects.
The goal of scTab is to do cell type classification, not batch correction.

Lastly, we refer back to the discussion for the reviewer's first point: it's clear that the training data
has flaws, and we've incorporated more discussion on this point in the manuscript. There are
indeed cell types and cell states that are recurrent throughout a spectrum of tissues and contexts
in the human body. If these annotations were named similarly and associated with the same term
in the Cell Ontology by data curators for CELLxGENE with permission from the contributing
authors, these cell types were merged together to be the same entity in the training data. That is
likely incorrect. It's likely if we associated the specific tissue context with the CL term, certain
predictions would be more accurate. However, we return to the fundamental flaw discussed in the
training data, which is to merge all cell annotations together as if the researchers are referring to
the same entity (when they are often not). We plan to return to these questions as we integrate
scTab into the Cell Annotation Platform, whereby this metadata is more systematically collected.

We have added the text below to our manuscript to briefly highlight the main idea behind modern
deep-learning models and that the goal of our paper is to see if these findings can be transferred to
cell type annotation of single-cell RNA-seq data:

Modern deep learning builds upon the idea of learning a decision function purely based on
data. This idea brought stunning breakthroughs in the fields of computer vision and natural
language processing, where the approach of using big models and large-scale training
datasets to regularize those models drastically outperforms other traditional (usually
feature-engineering-based) machine learning approaches. Hence, we leverage well-defined
benchmark metrics^{2,3} for cell type classification to understand the performance of deep
learning models trained on large scRNA-seq data corpora, focusing on the scaling behavior
of such models with respect to the training data size as well as the model size.

Moreover, we would like to refer to Fig. 1f, Supp. Figure 4 and the text below where we highlight for
the reader that the cross-entropy loss results in a more structured (batch-corrected) latent space
compared to the raw feature space (normalized to 10,000 counts and log_{1p} transformed):

When qualitatively inspecting the representations learned by scTab in a tSNE plot, we
found that cell types show consistent separation and that the latent space was more
structured compared to the raw feature space (Fig. 1f). Moreover, similar cell types group
together in the latent space (Supp. Fig. 3).

Lastly, we would like to refer to the changes from Question #1 about the limitations of the
CELLxGENE training data.

**4. (Line 156) The authors have set a threshold to remove rare cells with fewer than 5000**
**instances. However, I believe this threshold may be too stringent. In the context of**
**small-scale supervised learning, 5000 instances represent a significant amount, and this**
**criterion could potentially exclude relevant data.**

**Author response:**

Thank you to the reviewer for this feedback, it's a useful point to clarify.

The threshold of 5000 instances is defined in a dataset-dependent way and should be adjusted
accordingly for small-scale datasets. To clarify, the threshold is applied to the whole CELLxGENE
data corpus and not to individual datasets. Thus, in our case, 5000 instances correspond to around
0.02% of the total training data, hence excluding only a small fraction of the available cells in the
training data. Moreover, once more data becomes available, those cell types can be added to
training data again.

Note that in our case, the absence of more precise rare cell labels in the training data would simply
result in more broad (albeit precise) predictions against new test data. If the live data contained
these rare cell types, we would prefer to err on the side of caution with such a threshold.
Furthermore, as shown in the plot below (which we discuss in the response to Reviewer #1, Point
#3), one can see that for specific cell types the classification performance starts to degrade
considerably starting at approx. 10,000 cells in available training data. (Note on the plot the cells
with F1 scores less than 0.4 and sample sizes 10⁴.) As we wanted to use a general criterion
when excluding cell types and not a criterion based on classification performance, we would argue
that the 5000 instance threshold makes sense in our case. Especially, as already mentioned
above, we wanted to use a conservative threshold to rather err on the side of caution. Moreover,
using a more conservative threshold helps to keep class imbalances in check, which is a
substantial issue when dealing with scRNA-seq data.

Note: The 5000 instance threshold is calculated across the full dataset. Hence, the available
 training data is even smaller as part of the cells are assigned to the validation and test sets.

**5. How does the model handle new cell types in the query data that are not present in the**
 **reference dataset? This aspect is crucial for assessing the model's applicability to diverse**
 **and evolving datasets.**

**Author response:**

Thank you for this suggestion, this is indeed an interesting point.

Our intuition behind this model is that broad classes of cell types are inherently easier to predict
 than more precise classes of cell types, given the inherent nature behind annotating cells with a
 “broad” cell annotation label vs a more “precise” cell annotation label (e.g “myeloblast” vs
 “activated (M1) macrophage”). Precise cell types within a given “broad” cell type class are more
 transcriptomically similar to one another. (Note: we explicate this in greater detail within the
 response to Review #1 Question #2)

Currently, the training data used contains all “broad” cell types within the Cell Ontology across a
 range of tissues. If a new “coarse” cell type not yet characterized by the Cell Ontology was not in
 the training data, this would logically affect the performance of the model. Therefore, we’ll assume
 the reviewer is referring to more precise cell types not yet characterized, which is a common use
 case as more cells and tissues are sequenced.

As we discuss in Review #1 Question #2, the predictions in this case will be more accurate as
 broad cell types.

To the uncertainty of these predictions, we used deep ensembles (Lakshminarayanan et al, 2017,
 <https://arxiv.org/abs/1612.01474>). Deep ensembles are commonly used to assess the uncertainty
 in predictions of neural networks. They are simple, yet achieve state-of-the-art results: one just
 averages the predicted probabilities across several networks that were independently trained (each
 with a different random initialization of the weights).

In the next step, to assess how well one can identify cell types that are not present in the training
data or cells with wrong predictions we split the CELLxGENE data into three parts:

• **Group 1: Correct Predictions**

Cell types that are present in the training data and which are predicted correctly by the
model (this serves as a reference group). This group is referenced as in-distribution (right
prediction) or simply as “Group 1” below.

• **Group 2: Incorrect Predictions**

Cell types that are present in the training data but which the model predicted wrongly to
assess how well wrong predictions can be distinguished from right predictions based on the
uncertainty scores. This group is referenced as in-distribution (wrong prediction) or simply
as “Group 2” below.

• **Group 3: Absent in Training Data**

Cell types that are not present in the training data to assess how well unknown cell types
can be identified based on the uncertainty scores provided by scTab. These are the cell
types that we excluded from the CELLxGENE training data because there were too few
observations present. This group is referenced as out-of-distribution or simply as “Group 3”
below.

Now, to understand the quality of uncertainty estimates, we want to assess how well Group 2 and
Group 3 (incorrect and absent) can be separated from reference Group 1 (correct). Note that the
separation between Group 1 and Group 2 (correct vs incorrect) measures how well the uncertainty
scores can be used to assess whether a model prediction can be trusted or not, and the separation
between Group 1 and Group 3 (correct vs absent) gives an estimate of how well the uncertainty
scores can be used to detect new/unseen cell types. The model uncertainty is defined as follows
(logits are the outputs of the last layer of the neural network):

$$\text{uncertainty} = 1. - \max(p)$$

$$p = \text{softmax}(\text{logits})$$

To get a first impression, one can look at the distribution of uncertainty scores conditioned on which
of the three groups a cell belongs to:

As expected, one can see that the uncertainty scores for Group 2 and Group 3 are usually a lot
higher than for Group 1.

Now, to provide a more mathematically rigorous benchmark, one can measure how well one can
distinguish Group 2 and Group 3 from the reference Group 1 based on the uncertainty scores
provided by the scTab model by looking at the area under the curve of the receiver operating
characteristic (ROC-AUC). A ROC-AUC score of 1.0 means that the groups can be perfectly
separated and a score of 0.5 means that there is no separation between the groups based on the
uncertainty scores. The above approach can also be used to assess how the quality of uncertainty
estimates improves with the number of models in the deep ensemble.

Looking at the results, one can see that for both cases our uncertainty estimates provide a useful
way to distinguish between the groups. Group 1 and Group 3 can be separated with an ROC-AUC
score of 0.782 and Group 1 and Group 2 can even be separated with an ROC-AUC score of 0.891.

Code (see “Evaluate uncertainty estimates” section):

https://github.com/theislab/scTab/blob/devel/notebooks/model_evaluation/classification-tabnet-ens

embl.ipynb

In practice, biologists and computational biologists can overlay the uncertainty estimates from
scTab alongside the predicted cell type labels and their defined clustering on a UMAP or tSNE
visualization of their data to hint at which predictions are associated with higher uncertainty and,
hence, should be investigated in more detail.

We’ve added the results of this analysis in the supplementary (see Methods section and Supp. Fig.
5) and clarified in the main manuscript for readers.

**6. The author describes a data augmentation procedure involving the shifting of**
**measurements across donors, learned from other cell types. For instance, the**

augmentation vector for T cells is captured using B cells. However, I am concerned about
the validity of this approach. The gene space might interact differently with various cell
types across donors, potentially affecting the reliability of the augmentation (because the
augmenting vector is influenced by cell type). Could the authors clarify how this method
accounts for the potential interaction between the cell types and donors in the change of
gene space?

**Author response:**

Thank you to the reviewer, this indeed deserves clarification regarding how we accounted for the
potential cell type dependency of augmentation vectors. However, before explaining this in more
detail we would first like to emphasize that, the usefulness of our data augmentation strategy
should be evaluated in terms of improved model generalizability (lower test/validation loss and/or
higher test/validation F1-score) as it gives a direct measure of what we want to achieve - that's
also the evaluation we have focused on in our paper. There's always a trade-off between artificially
increasing the training data size through data augmentation (the augmented data points are
artificially created and thus come with some approximation error) and avoiding overfitting (more
data helps to reduce overfitting); looking at the improvement in model generalizability gives a
measure of this and whether the augmentation scheme works.

To get an impression of how strong the cell type dependency of the data augmentation vectors is,
we first look at a tSNE plot of the data augmentation vectors with the cell type based on which the
augmentation vector was calculated superimposed:

One can see evidence of distinctly cell type-specific clusters in the tSNE plot (e.g. 'fibroblast of
cardiac tissue', 'epithelial cell of proximal tubule', 'glutamatergic neuron', etc.) This means that
those augmentation vectors are cell type dependent as the reviewer pointed out. However, there
are also augmentation vectors where this cell type dependency is a lot weaker (meaning the
vectors do not cluster by cell type) as can be seen in the middle of the tSNE plot (e.g. 'fallopian
tube secretory epithelial cell', 'alveolar macrophage', etc.).

Now, to ensure that augmentation vectors can indeed be applied in a cell-type-independent
fashion, we want only to select augmentation vectors that are mostly cell-type-independent. To do
so, we clustered the augmentation vectors via K-means clustering. After clustering, one can
observe that the resulting clusters are quite different in size, meaning some contain many
augmentation vectors and others contain far fewer augmentation vectors. More importantly, we see
that the k-means clusters with many augmentation vectors are mostly cell type independent i.e.

they do not cluster by cell type. Hence, as described in the methods section of the paper, we only
 selected augmentation vectors from the biggest K-means clusters. Now, looking at the tSNE plot of
 the data augmentation vectors we used for model training (after the K-means filtering step was
 applied), one can see that the above-described filtering technique indeed results in mostly cell type
 independent augmentation vectors.

 We've added more clarifying details to the methods section of the paper.

 **7. (Line 193) While I appreciate the presentation of the standard deviation in the results, the**
 **method of its calculation remains unclear. The small magnitude of the standard deviation**
 **values is particularly surprising. Could the authors provide more details on how these**
 **values were computed? Specifically, does the repetition of the experiment involve**
 **bootstrapping of the sample?**

 **Author response:**
 We thank the reviewer for pointing this out. We calculated the standard deviations across several
 model fits (fitted with a differently seeded random initialization) but using the same evaluation data.
 That's probably why the confidence intervals are so low. However, the reviewer has a valid point.
 So, we additionally provide standard deviations which include bootstrapping across samples
 (donors) in the table below:

	F1-score (macro avg.)	Number of runs to calculate standard deviation
scTab (deep learning)	0.8300 ± 0.0069	4
XGBoost (boosted decision trees)	0.8136 ± 0.0060	4
MLP (deep learning)	0.7973 ± 0.0074	4
Linear	0.7846 ± 0.0072	4
CellTypist (training data subsampled to 1.5 Mio cells)	0.7291 ± 0.0072	4

 We've consequently added a clarification to the manuscript and additionally added the above table
 to the supplements (Supp. Tab. 1b).

**8. (Lines 205-210) The comparisons made with existing methods appear to be potentially**
**unfair. For example, the XGBoost model was not fitted with data augmentation, unlike the**
**proposed method. Similarly, scGPT was used without fine-tuning. These discrepancies**
**might lead to a skewed assessment of the performance of the proposed method against**
**these established techniques.**

**Author response:**

Thank you for the questions. For organization's sake, we will respond first to the point regarding
XGBoost, then address the point regarding scGPT, and then address the final point above.

**> For example, the XGBoost model was not fitted with data augmentation, unlike the**
**proposed method.**

To answer this question, we would first like to provide a general overview of data augmentation
strategies in machine learning. In general, one might classify data augmentation strategies into two
groups. The first group consists of easy-to-apply transformations that can be applied on the fly
before feeding the data into for example a neural network. This strategy is very common in
computer vision, where one can e.g. rotate, crop, or flip an image before feeding it into a neural
network. This kind of strategy requires that the model is trained in mini-batches; meaning the
transformation can be applied to each mini-batch before using it as input to a neural network. The
second group of data augmentation strategies consists of more compute-heavy augmentation
strategies, which can be used to extend/augment existing datasets. This for example contains tools
like SMOTE or using generative models to generate new training samples. As those data
augmentation strategies are a lot more compute-heavy, they cannot be applied on the fly to a
mini-batch anymore as this would otherwise significantly slow down the training. Now, with our
augmentation strategy, we wanted to have a computer vision-inspired augmentation strategy, e.g. a
strategy that is applied on the fly to each mini-batch. Similar to computer vision, our model is fitted
for several epochs; meaning by applying those on-the-fly data transformations, the model never
sees the same data point twice. Hence, avoiding overfitting.

Now, to explain why we did not use XGBoost with our data augmentation strategy. Unlike our
scTab or the MLP model from our paper, XGBoost is not fitted in mini-batches. Thus, it's not
straightforward to apply our data augmentation strategy in combination with XGBoost. Moreover,
there are also more practical reasons that currently make it questionable, whether it makes sense
to try to use XGBoost in combination with data augmentation in general in our use case. Firstly,
XGBoost already struggles to scale to the current dataset size available to us, as it keeps all data
in memory (it's not fitted in mini-batches) - we already used the 80GB version of the A100 GPU for
XGBoost to be able to fit the model without sub-sampling the data (all other models were trained
on the 40GB version). Moreover, again due to memory and runtime constraints, we had to fit the
XGBoost model on 256-dimensional PCA embeddings, this further complicates the application of
our data augmentation strategy as it is designed to be applied in the raw feature space and not in
the PCA space.

**> scGPT was used without fine-tuning.**

We fine-tuned a scGPT model on a random subsample of 150,000 cells of our training data and
reported the macro and micro averaged F1-Scores in the table below:

	F1-Score (macro avg.)	F1-Score (micro avg.)
scGPT fine-tuned with 150,000 cells	0.749	0.877

The fine-tuned scGPT model indeed performs better than just using the scGPT embeddings in the
zero-shot setting, where scGPT achieved a macro F1-Score of 0.73. However, at this point, we
would like to highlight a few potential issues we encountered with the scGPT model. The biggest
issue with scGPT or related single-cell foundation models like GeneFormer or Universal Cell
Embeddings is their immense computational costs. Those models not only have significantly higher
compute requirements for model training but also for model inference. Annotating the 3.5 million
cells from our test set took several hours with the scGPT model, whereas all other models from our
paper only needed several minutes. Given that one can achieve similar or better performance with
smaller models, it becomes hard to justify the significantly increased computational costs.
Moreover, currently, the scGPT code base does not have a stable API for model fine-tuning but
only a (somewhat) unstructured example of how to fine-tune a scGPT model for cell type
classification in a Jupyter notebook (https://scgpt.readthedocs.io/en/latest/tutorial_annotation.html).
Given this information isn't easily accessible, it's a great investment in time to figure out how to
fine-tune a scGPT model on custom data; this was also the reason why initially did not include a
fine-tuned scGPT model in our benchmarks. Lastly, based on the code in the above-mentioned
notebook, scGPT has quite high memory requirements. Fine-tuning the scGPT model on only
150,000 cells already required 100+GB of memory. Therefore, we were forced to fine-tune the
model on a rather small subsample of our training data. As we mentioned above, despite only
using 150,000 cells, the fine-tuning already took several hours, requiring a similar amount of
computing resources as the other models in our paper.

> **These discrepancies might lead to a skewed assessment of the performance of the**
**proposed method against these established techniques.**

This is a fair point, given the reasons above. We addressed this concern by a note of clarification
regarding XGBoost in the comparison, and by adding a fine-tuned scGPT model to our
benchmarks.

**9. (Line 269) Could the authors provide the p-value associated with the reported correlation**
**of -0.55?**

**Author response:**

We thank the reviewer for spotting the missing p-value. The value of the test statistic and the
associated p-value can be seen below:

`PearsonRRResult(statistic=-0.5463034007214954, pvalue=0.0351169140580146)`

We have consequently updated the main manuscript with this clarification.

**10. (Line 562) The authors have chosen the macro F1-score as an evaluation metric, which,**
**as noted in the discussion, could be influenced by class size imbalance. It may be**
**beneficial for the authors to also consider presenting the micro F1-score.**

**Author response:**

Thank you to the reviewer for this feedback, it's a useful point to clarify.

We have chosen the macro average as it is much more robust concerning class imbalances as it
gives every cell type the same weight in the final score. That is, each cell type has the same weight
independent of the number of cells for each cell type. On the other hand, the micro-average is
strongly influenced by class imbalances, meaning cell types that are overrepresented in the
evaluation data are also weighted more strongly in the final score. As we want to assess the
classification independent of class imbalances, we chose the macro average. However, we
additionally listed the micro-averaged F1-Scores in the table below:

	F1-score (micro avg.)	Number of runs to calculate standard deviation
scTab (deep learning)	0.9067 ± 0.0066	4
XGBoost (boosted decision trees)	0.8977 ± 0.0057	4
MLP (deep learning)	0.8824 ± 0.0067	4
Linear	0.8700 ± 0.0061	4
CellTypist (training data subsampled to 1.5 Mio cells)	0.8487 ± 0.0088	4

For more details regarding the calculation of micro and macro average F1-Scores, see below:

- ● Micro average:
 - ○ **Calculate metrics globally by counting the total true positives, false negatives**
 - **and false positives.**
 - ○ Hence, it is affected by class imbalances as bigger classes contribute more samples
 - to the calculation of the F1-Score than smaller underrepresented classes, which
 - consequently gives overrepresented classes a bigger weight in the overall
 - F1-Score. Which, in our case, is not desired as we want each cell type to be
 - represented equally.
- ● Macro average:
 - ○ **Calculate metrics for each label, and find their unweighted mean. This does**
 - **not take label imbalance into account.**
 - ○ Hence, is robust to class imbalances as each class label has the same contribution
 - to the overall F1-Score. This is the behavior we want in our case, where the final
 - F1-Score is influenced by each cell type in the same way, independent of class
 - imbalances.

The bold highlighted sentences are taken from the scikit-learn documentation:

https://scikit-learn.org/stable/modules/generated/sklearn.metrics.f1_score.html

Moreover, we created a simple example to showcase the advantage of the macro-averaged
F1-score over the micro-averaged F1-score when dealing with imbalanced classification problems.
Let's assume we have a classification problem with three classes: Class 0 with 1000 examples and
class 1 and 2 with 50 examples each. By definition, this is an imbalanced classification problem.
Now, let's assume we have a model that always predicts the majority class 0. One can see from
the metrics calculated below that the macro F1-score gives a far more accurate estimate of the
classification performance given that we equally care about the classification performance of each
class:

```
[1]: from sklearn.metrics import accuracy_score, f1_score
```

```
[ ]:
```

```
[2]: y_true = [0] * 1000 + [1] * 50 + [2] * 50  
y_pred = [0] * 1100
```

```
[3]: f1_score(y_true, y_pred, average='macro')
```

```
[3]: 0.31746031746031744
```

```
[4]: f1_score(y_true, y_pred, average='micro')
```

```
[4]: 0.9090909090909091
```

```
[5]: accuracy_score(y_true, y_pred)
```

```
[5]: 0.9090909090909091
```

On another note, in the multi-class setting and when all labels are included in the evaluation, the
micro-averaged F1-score is equivalent to the accuracy. See scikit-learn documentation for more
details:

https://scikit-learn.org/stable/modules/model_evaluation.html#multiclass-and-multilabel-classificatio
n

In summary, we provided a brief summary of our rationale using the macro F1-Score in the
manuscript, and we hope we were able to clarify the advantages of the macro F1-score over the
micro F1-score to the reviewer with the more detailed response above.

**11. (line 591) This sentence is confusing. A revision might be needed:" On the one hand, the**
**feature transformer is used to get a lower dimensional representation (described by**
**dimensionality nd) which is used to classify cell types. On the other hand, it is used to get a**
**lower dimensional representation (described by dimensionality na) from which the feature**
**attention mask is calculated. "**

**Author response:**

We thank the reviewer for pointing this out. We updated the manuscript with an updated version of
the sentence:

The feature transformer maps from the input gene expression space to an $n_d + n_a$
dimensional latent space. In the next step, the $n_d + n_a$ dimensional embedding is split
into two parts: one with dimension n_d and one with dimension n_a . The part with
dimension n_d is used to classify the different cell types and the second part with
dimension n_a is used to calculate the attention masks.

*Minor Comments:*

**12. (line 357) Typo "datase"**

**Author response:**

Thank you, we've corrected this typo

**13. (line 600) Notations "p" and "\$p_j\$" are undefined.**

**Author response:**

Thank you, we've clarified this notation in the manuscript

**Reviewer #4**

This study establishes a comprehensive dataset designed to facilitate the training and
evaluation of a cross-tissue cell type classification model. This initiative represents a
substantial advancement in the landscape of cell type identification, making noteworthy
contributions to the field. Additionally, the paper introduces scTab, an automated cell type
prediction model meticulously tailored for tabular data, employing a feature-attention-based
methodology. The model undergoes training using an innovative data augmentation
scheme applied across an extensive repository of single-cell RNA-seq observations,
encompassing a total of 22.2 million human cells. In summary, this paper holds substantial
potential for impact, supported by its anticipated high citation rate. The overall presentation
of the paper is commendable and warrants publication with only minor revisions.

**Author response:**

Thank you for the summary, kind evaluation, and suggestions!

Several queries arising from the content have proven perplexing, and your insights would
be greatly appreciated:

**1. I suggest using specific model names to differentiate between the "linear (ours) model"**
**and a generic "linear model."**

**Author response:**

We thank the reviewer for this suggestion, we adapted the suggestion "optimized linear" from
Reviewer # 1 and updated the manuscript accordingly.

This did indeed need clarification.

**2. Could you please provide a clear definition for the "cross-tissue annotation task"?**
**Initially, I inferred that the model was trained on data from one tissue and tested on another.**
**However, upon careful examination of the paper, it became evident that this interpretation**
**was not accurate.**

**Author response:**

We thank the reviewer for the suggestion; indeed, this should be clarified in more detail. By
"cross-tissue annotation task" we mean training and evaluating a single classifier model on a
diverse selection of tissues. This is in contrast to organ-specific classifiers which are trained and
evaluated only on a single tissue or only a narrow selection of closely related tissues.

We have clarified this in the main manuscript:

In this context, by "cross-tissue annotation task" we mean training and evaluating a single
classifier model on a diverse selection of tissues. This is in contrast to organ-specific
classifiers which are trained and evaluated only on a single tissue or only a narrow
selection of closely related tissues.

**3. Clarification is sought regarding any distinctions between "cross-tissue annotation" and**
**"general cell type annotation."**

**Author response:**
We thank the reviewer for pointing this out. This deserves clarification. With both "cross-tissue
annotation" and "general cell type annotation", we essentially refer to the same task; namely,
directly predicting cell type labels solely based on the gene expression of a cell with a single
unified model that works across a wide range of tissues. However, this task is not fully solved yet
as the data and the quality of cell type annotations are just not there yet. To highlight this we used
two different terms: "Cross-tissue annotation" refers to the general task; "General cell type
annotation" refers to the overall goal we want to achieve - one might call it the "holy grail" of cell
type annotation.

We have accordingly added these clarifying details in the main manuscript to avoid confusion:
Here, we would like to add a clarification note regarding the distinction between
cross-tissue and general cell type classification: with both terms, we essentially refer to the
same task; namely, directly predicting cell type labels solely based on the gene expression
of a cell with a single unified model that works across a wide range of tissues. However,
this task is not fully solved yet as the data and the quality of cell type annotations are just
not there yet. To highlight this we used two different terms: "Cross-tissue annotation" refers
to the general task; "General cell type annotation" refers to the overall goal we want to
achieve - one might call it the "holy grail" of cell type annotation.

**4. The decision to reject sequence-based models, such as Transformers, due to the**
**unordered nature of gene expression profiles is reasonable. Instead, the authors opted for a**
**recent architecture explicitly designed for tabular data, namely TabNet. Additionally, scGPT**
**incorporates a novel attention mechanism to address this challenge. While it is**
**conceptually reasonable that sequence models may not be ideal for non-sequential data,**
**uncertainty remains regarding whether the unordered nature of gene expression profiles**
**genuinely impacts the performance of Transformer-based models. Some Transformer-based**
**models employed for cell type identification, such as CIFORM [1] and TOSICA [2], despite**
**overlooking the unordered nature of genes, have demonstrated commendable performance.**
**I recommend comparing this aspect with TOSICA and CIFORM. If a direct comparison is**
**considered inappropriate, please provide a suitable rationale.**

**Author response:**
We thank the reviewer for this suggestion. The reason why we didn't add the two models in our
initial benchmark is that both models do not scale to big datasets. Both models load the full training
data into memory in a dense format, which comes with high memory usage. Thus, limiting the
applicability of both models to only relatively small datasets. In our paper, we investigate how cell
annotation models scale to large-scale datasets, which due to the above reason, is not possible for
either of the two models. However, to provide a performance estimate, we ran the CIFORM model
on a subsample of 750,000 cells and reported the model performance below. As expected due to
the limited training data compared to our other models, the model performance is quite a bit lower.
Moreover, we tried running the TOSICA model as well, which on the one hand had extensive
memory requirements despite only being fitted on a subset of 100,000 cells and, additionally,

suffered from a GPU memory leak, making it impossible to fit in our case as the model quickly ran
out of GPU memory during model training. People have already reported this behavior on their
GitHub repository: <https://github.com/JackieHanLab/TOSICA/issues/16>. Unfortunately, the issue is
unaddressed by the authors of the paper.

The CIFORM model achieved a macro F1-score of 0.7660. Additionally, for better comparison, we
plotted the classification performance of our scTab model, the optimized linear model, the
CellTypist model, and the CIFORM model in the barplot below:

We've added the CIFORM model to our benchmarks in the manuscript.

**5. The paper introduces scTab as a tool for cell type annotation. I am intrigued by its**
**performance on widely used benchmarking datasets, such as the Zhang68K [3] dataset.**

**Author response:**

We thank the reviewer for the suggestion. We used the pbmc68K dataset from the scVelo package
for our evaluations (<https://scvelo.readthedocs.io/en/stable/scvelo.datasets.pbmc68k.html>), as the
dataset provided in the original paper did not include any cell type annotations. Furthermore, we
split the data according to a 70-15-15 train, validation, and test split and reported the macro and
micro averaged F1-Score on the test set in the table below:

	F1-Score (macro avg.)	F1-Score (micro avg.)
scTab	0.484	0.617

We would like to emphasize that we intended scTab to be used in combination with a large-scale
training corpus. One key point of our paper is to highlight the potential of using a large-scale
training data collection consisting of many individual datasets in combination with larger more
expressive models - similar to what has been shown in the AlexNet paper (Krizhevsky et al.) for

computer vision. However, larger models like scTab are inevitably prone to overfitting on
small-scale datasets like the Zhang68K dataset, which is over a factor of 300 smaller than the
training data we used to train our scTab model. Thus, the Zhang68K dataset might not be the ideal
setting to evaluate the potential of our scTab model.

We have added the benchmark results to Supp. Table 11 of the paper.

References:

1. Xu J, Zhang A, Liu F et al. CIFORM as a Transformer-based model for cell-type annotation of
large-scale single-cell RNA-seq data, *Brief Bioinform* 2023;bbad195.

2. Chen J, Xu H, Tao W et al. Transformer for one stop interpretable cell type annotation, *Nat*
*Commun* 2023;14:223.

3. Zheng GX, Terry JM, Belgrader P et al. Massively parallel digital transcriptional profiling of single
cells, *Nat Commun* 2017;8:14049.

Reviewer #1 (Remarks to the Author):

We thank the authors for the extensive response to our review comments. We think they have addressed most of our points, however, there are still a few minor issues that we describe below.

Major points - Round 2 comments

1. We thank the authors for evaluating the improvement in the celltype annotations by scTab and for the detailed response provided. Can we please confirm if the top25 differences in F1-score reported in Supplementary Fig.7b always correspond to better performance in scTab?

The authors show that the improvement in prediction occurs in more fine-grained cell types, like T cell lymphocytes, where the ontology provides more detail in terms of cell subpopulations. It would be ideal to provide a comprehensive picture of how this looks for other cellular compartments. Could the authors provide a summary of how many individual labels contribute to a particular cell lineage? They specify how this looks for T cells but ideally, they would report this in a systematic way. Following from the previous question, an assessment of how the performance improves with respect to the optimized linear model in relation to the level of granularity for each cell category in the hierarchy, should be ideally provided to fully convey the value of scTab. Having said this, we think that the redundancies between annotation labels due to lack of label harmonization across datasets could contribute to the improvement of performance in a way that is difficult to judge and this should be clearly stated. For example, in figure suppl 7 we can see how very general labels such as lymphocyte or ab-T cells contribute to the improvement of performance.

2. We thank the authors for their response. Regarding the granularity of cell type annotations, authors responded above how they filtered for cell types with at least seven parent nodes in the ontology graph. It seems that still after this filtering step, there are very general terms such as lymphocyte included in the high granularity predictions. This is understandable given that curating and harmonizing labels is an extremely time-consuming task, however, this should be clearly explained as a potential limitation of the pre-trained models as they stand now.

3. We thank the authors for their detailed response and for making clear that there are specific cell type labels with higher classification performance while having lower counts. We would suggest including an r-square value in the regression in Figure 1g. to evaluate the correlation between abundance and classification.

4. We thank the authors for addressing this point. It seems like the new Suppl. Table 8 is not referenced in the manuscript.

5. We thank the authors for doing a more general comparison of the performance across different organs vs cross-tissue training of the model. We agree that scTab shows a more robust behaviour in the cross-tissue context, however, we consider that the lack of label harmonization across tissues could be a factor confounding this result and this should be stated. Also, could the authors please clarify why there is no error bar for specific tissues in the optimized linear bars?

6. Thanks for the response. We refer to the answer to point 5 on acknowledging the potential effect of lack of label harmonization on the robustness of the scTab model across tissues, e.g. one of the labels with robust annotation in scTab but not in the optimized linear model is lymphocyte, however, given that this is a very general label, could it be that obtaining a prediction matching more refined labels from a different tissue could be an equally correct or even more helpful result? We understand this requires the time-consuming label harmonization that is not on the scope of this study but consider that it is important to mention it.

7. We thank the authors for making available the pre-trained models. In connection to the point n.5 from reviewer 4, we would suggest including an application of the pre-trained model(s) to real use-case datasets. To this end, we would suggest providing an example tutorial with a benchmark dataset where also the confidence metrics, such as the above mentioned F1 score, can be visualized. In addition, the application of these pre-trained models to annotate specific disease states, as the dataset employed for the training includes 52 disease states, would be of interest to

the community.

8. Thanks to the authors. We refer to our response to point 7.

9. We thank the authors for the reasoning of this point. We agree with the response provided.

10. We thank the authors for addressing our concern

Minor points - Round 2 comments

1. We thank the authors for their response. We would suggest including more detail on how cell type labels are distributed across tissues and how they are related in the hierarchy, also, in connection to major point 1.

2. We thank the authors for this, however, there is still one occurrence in line 786, we ask the authors to please modify.

All other minor points were addressed.

Reviewer #2 (Remarks to the Author):

Reviewer #3 (Remarks to the Author):

Please see my comments in the attachment.

Reviewer #3 Attachment on the following page

Referee Report for NCOMMS-23-55310A

Scaling cross-tissue single-cell annotation models

The authors have addressed most of my comments and concerns. The clarification is very helpful. Stemming from the discussion of the response letter, I have some clarification questions. Hope it can be helpful:

1. (From #5) The author mentioned a very interesting point regarding using uncertainty scores to discriminate between correct predictions and the incorrect or newer group. I was wondering if there's any specific guidance for users regarding how to use or interpret these uncertainty scores to do such discrimination.
2. (From #6) The author mentions in method section a filtering step for selecting a more focused group of augmentation vectors to ensure they are insensitive to the cell types. I was wondering if the authors could clarify with more details about how such a procedure is performed so that the users can perform in practice. For example, how K is determined such a K-means clustering?

Reviewer #4 (Remarks to the Author):

The response to my review reflects a thoughtful consideration of feedback, resulting in substantial enhancements to the manuscript. The authors' revisions significantly bolstered clarity and effectively addressed key concerns. The paper introduces scTab, a novel approach for cell type prediction in single-cell transcriptomics. I recommend acceptance for publication based on the manuscript's improved quality and its potential to propel cell type annotation in the field of single-cell transcriptomics.

Reviewer #1 (Remarks to the Author):

We thank the authors for the extensive response to our review comments. We think they have
addressed most of our points, however, there are still a few minor issues that we describe below.

Author response:

*Major points - Round 2 comments*

**1. We thank the authors for evaluating the improvement in the celltype annotations by
scTab and for the detailed response provided.**

Thank you for the kind comments and suggestions. For the sake of organization, we split our
response into several parts:

**> Can we please confirm if the top25 differences in F1-score reported in Supplementary
Fig.7b always correspond to better performance in scTab?**

Yes, all the top 25 differences correspond to better performance in scTab. We defined the
difference as follows:

$$15 \quad F1_{Difference} = F1_{scTab} - F1_{Optimized linear}$$

Thus, a positive difference corresponds to better performance in scTab. And all differences are
positive as one can see from the plot.

**> The authors show that the improvement in prediction occurs in more fine-grained cell
types, like T cell lymphocytes, where the ontology provides more detail in terms of cell
subpopulations. It would be ideal to provide a comprehensive picture of how this looks for
other cellular compartments. Could the authors provide a summary of how many individual
labels contribute to a particular cell lineage? They specify how this looks for T cells but
ideally, they would report this in a systematic way.**

As it is difficult to define good coarse cell type groupings based on the Cell Ontology, we would first
like to provide more examples related to the immune system (namely using T cells, B cells,
monocytes, macrophages, granulocytes and dendritic cells as coarse labels). And, thereafter
present a more systematic overview across cell lineages based on coarse cell type defined by the
"cell subclass" field from CELLxGENE.

*More detailed look at cells of the immune system:*

Looking at the plot below, one can see that there are similar trends for other cell types of the
immune system, namely: T cells, B cells, monocytes, macrophages, granulocytes, and dendritic
cells. Notably, the trend is strongest for T cells, which is in line with the observation that T cells are
by far the most prominent cell types in the CELLxGENE data corpus (37 subtypes for T cells, 9
subtypes for B cells, 8 subtypes for monocytes, 8 subtypes for macrophages, 2 subtypes for
granulocytes and 4 subtypes for dendritic cells). Interestingly, the trend is even visible for
granulocytes for which only two subtypes are present in our data corpus.

 **Supp. Fig. 8 (a):** Evaluation on selected coarse cell type labels of the immune system: T cells, B cells, monocytes,
 macrophages, granulocytes, and dendritic cells. The number of fine-grained cell type labels corresponding to a coarse
 cell type label is indicated beside the cell type name.

 *Systematic evaluation:*

To provide a systematic overview, we selected coarse cell type labels as defined by CELLxGENE
 in the “Cell Subclass” field (those labels can be e.g. seen here:
 [https://chanzuckerberg.github.io/cellxgene-census/notebooks/analysis_demo/comp_bio_geneform](https://chanzuckerberg.github.io/cellxgene-census/notebooks/analysis_demo/comp_bio_geneformer_prediction.html#Preparing-data-from-model)
 [er_prediction.html#Preparing-data-from-model](https://chanzuckerberg.github.io/cellxgene-census/notebooks/analysis_demo/comp_bio_geneformer_prediction.html#Preparing-data-from-model)). As this coarse label set still includes 88 labels, we
 further subsetted this set with the following criteria to aid visualization:

- 1. The coarse label needs to have at least 7 subclasses in our label set
 2. The coarse label should have less than 50 subclasses in our label set to avoid too coarse
 label sets

Based on this filtering criterion, we ended up with a total of 26 coarse cell type labels.

Now, to provide a systematic overview, we plotted the difference in F1-score between the scTab
 model and the optimized linear model on the coarse labels plotted on the x-axis and the difference
 in F1-score on the fine-grained annotations plotted on the y-axis in the barplot below. Additionally,
 we added a red reference line: For all the points above the red reference line, the performance
 improvement of scTab compared to the optimized linear model increased when evaluating the
 classification performance on the fine-grained cell type labels.

**Supp. Fig. 8 (b):** Systematic evaluation on cell types across the CELLxGENE data corpus. The plot shows the difference
 in F1-score between the scTab model and the optimized linear model on the coarse labels plotted on the x-axis and the
 difference in F1-score on the fine-grained annotations plotted on the y-axis in the barplot below. Additionally, shows a red
 reference line: For all the points above the red reference line, the performance improvement of scTab compared to the
 optimized linear model increased when evaluating the classification performance on the fine-grained cell type labels.

From the plot above, one can see a similar trend for other cell lineages as well. (Note there are
 exceptions where both scTab and the optimized linear model achieve a similar performance
 regardless of the label granularity e.g. brain-related cell types, secretory cells or contractile cells).
 One possible explanation for this phenomenon is that the optimized linear model already achieves
 a high classification performance when distinguishing these subtypes; thus, the room for
 improvement is limited. This can be seen in more detail in the barplot below where we show the
 number of unique CL terms per coarse cell type labels and additionally the classification
 performance of the optimized linear model for distinguishing the fine-grained cell type labels
 associated with the respective coarse label (measured by macro F1-score). From the plot one can
 see that for some cell types the optimized linear model already achieves a considerable
 classification performance despite there being a decent amount of fine-grained labels associated
 with the coarse cell type label. Some noteworthy examples for this are neural cells and secretory
 cells. This again highlights that measuring the complexity of distinguishing subtypes by the number
 of child nodes in the Cell Ontology; relying upon the count of child nodes per CL term is only a
 heuristic, and that the true complexity of the data is far more complex to measure. This however is
 beyond the scope of this paper and could be a topic for future work.

We've updated the manuscript text in the following way to reflect the reviewer's request:

Now, looking in more detail where the performance improvement of scTab stems from, one
 can observe that scTab performs particularly well in distinguishing between closely related
 classes of refined cell subtypes. This is most prominent when looking at T cell subtypes
 (Supp. Fig. 7b) for which the Cell Ontology is especially detailed, thus making it possible to
 resolve such fine-grained differences between subtypes. Notably, this trend is in fact not
 only limited to T cells but can also be observed in other cell lineages of the immune system,
 namely, B cells, monocytes, macrophages, and granulocytes. (Indeed, for many immune
 cell lineages, the Cell Ontology is quite detailed, and therefore these subtypes existed in
 the training data.) To assess the above trend quantitatively, for both scTab and the
 optimized linear model we compared the classification performance of identifying cells
 associated with a specific cell lineage against the classification performance of
 distinguishing fine-grained subtypes associated with the respective cell lineage. Comparing
 the difference in classification performance of the two models for those two settings, one
 can see that both models achieve similar performance when identifying cells of a particular

cell lineage but that the performance difference becomes more pronounced when
distinguishing fine-grained subtypes for scTab (Supp. Fig. 8a). This example highlights the
potential of our scTab model to distinguish between closely related cell types and hints that
the improvement of non-linear models such as scTab will further increase once more finely
annotated training data becomes available, and the Cell Ontology incorporates more
finely-grained subtypes of cells in other tissues. It is perhaps unsurprising how relatively
detailed immune cells are in the Cell Ontology with respect to other systems. Moreover, this
trend does not seem to be limited to only the immune system but can be observed in other
cell lineages outside of the immune system as well (Supp. Fig. 8b). Nonetheless, we would
like to emphasize here that this trend does not hold in general for all tissues. Cell types
related to the brain are the most notable exception here, which we hypothesize is probably
influenced by the relative lack of resolution of the Cell Ontology for brain-related cell types.

> **Following from the previous question, an assessment of how the performance improves**
**with respect to the optimized linear model in relation to the level of granularity for each cell**
**category in the hierarchy, should be ideally provided to fully convey the value of scTab.**

Associating the prediction performance with the granularity of cell type labels is not straightforward
for several reasons. Firstly, it's difficult to accurately measure the granularity of a CL term. We
discuss how the Cell Ontology is a work in progress in response to Major Point #2 below. Given
how the Cell Ontology is constructed, there simply isn't a clear relationship between the number of
leaf nodes of the CL hierarchy and the "granularity" of the term being associated with a specific cell
type or cell state. The Cell Ontology varies drastically based on the research priorities at the time,
and certain tissue systems are simply more represented than others. (Hence, a heuristic to reliably
filter out coarse cell types will always be complicated; see more to our response in Major Point
#2.). Secondly, just because a cell has a granular label in CELLxGENE doesn't mean it's easy to
predict. As we detailed, upon submitting datasets to CELLxGENE, one set of author provided cell
annotations are mapped to these author annotations as follows:

*To meet CELLxGENE schema requirements, the most accurate available CL term*
*MUST be used until the new term is available. For example if*
*cell_type_ontology_term_id describes a relay interneuron, but the most accurate*
*available term in the CL ontology is CL:0000099 for interneuron, then the*
*interneuron term can be used to fulfill this requirement and ensures that users*
*searching for "neuron" are able to find these data. If no appropriate term can be*
*found (e.g. the cell type is unknown), then "unknown" MUST be used. Users will still*
*be able to access more specific cell type annotations that have been submitted with*
*the dataset (but aren't required by the schema).*

Please refer to the latest schema v5.1.0 for these details:

<https://github.com/chanzuckerberg/single-cell-curation/blob/main/schema/5.1.0/schema.md>

There are in fact many cases where a very fine-grained author annotation gets mapped to a coarse
CL term. A notable example relates to certain regions of the human brain, where the Cell Ontology
remains extremely limited. Here, most fine-grained cell types provided by the authors can only map
to very broad "neuron" cell types. Another common example is when authors provide broad cell
annotations to cells which are not directly related to their study at hand; these broad cell labels

provided by the author will logically map to very broad CL terms. Therefore, the assumption that
coarse labels are easy to predict does not reflect the data well.

In summary, we would like to emphasize that we find it conceptually more clear to group together
related cell type labels together to form more coarse cell type labels (like we did in our response
above) as this better reflects the notion of a coarse cell type as biologists would define this when
annotating broader classes of cell types (i.e. larger clusters and therefore less precise terms)
instead of relying on the often wrong assumption that a coarse cell type label in CELLxGENE
corresponds to a more broad cell type definition. This is the primary reason why such a
performance assessment wasn't presented in earlier drafts of the manuscript.

> **Having said this, we think that the redundancies between annotation labels due to lack of
label harmonization across datasets could contribute to the improvement of performance in
a way that is difficult to judge and this should be clearly stated. For example, in figure suppl
7 we can see how very general labels such as lymphocyte or ab-T cells contribute to the
improvement of performance.**

As we mentioned, the training data is quite limited and does contain errors. The assumption that
authors would uniformly associate molecular signatures with the most precise Cell Ontology terms
is flawed. This was a major topic of revisions during the first round of reviewer responses, and we
tried to modify the manuscript accordingly in order to emphasize this fact. The community engages
in a great deal of manual curation to resolve inconsistencies and errors within the cell annotations
themselves. We suspect this work will easily take several years.

One promising avenue for comprehensive references of the identity of cell types and cell states are
community-driven reference atlases constructed by the HCA (and other efforts). We hope this will
provide more accurate training data for classification tasks in the future.

**2. We thank the authors for their response. Regarding the granularity of cell type
annotations, authors responded above how they filtered for cell types with at least seven
parent nodes in the ontology graph. It seems that still after this filtering step, there are very
general terms such as lymphocyte included in the high granularity predictions. This is
understandable given that curating and harmonizing labels is an extremely time-consuming
task, however, this should be clearly explained as a potential limitation of the pre-trained
models as they stand now.**

We agree with the reviewer that this should be explained in more detail. We added the following
clarifying details:

Added clarifying details when explaining the label selection heuristic:

Here, we would like to emphasize that the previously described filtering step is merely a
heuristic, meaning that there are still coarse cell type labels present in the scTab data
corpus even after our filtering heuristic.

Added clarifying details in the discussion section:

We would also like to highlight once more that the current pre-trained scTab models still
contain some fairly coarsely annotated cell types despite our efforts to filter out too coarsely
annotated cell type labels.

In terms of why this heuristic does not always result in more granular terms, this is largely a reality
of how the Cell Ontology is curated. The Cell Ontology grows based on funding associated with
certain tissues and consortia. These areas of the Cell Ontology receive more attention from
curators, and the terms are often more refined. Usually, the manner in which new terms are added
to the Cell Ontology is based on new scientific publications with novel terms. Ontology curators at
EMBL-EBI review those publications and add new terms to the ontology. By that very fact, some
tissues are better characterized than others given research interests and funding. Given this reality,
there was no heuristic available for us to filter for “granular terms only”.

**3. We thank the authors for their detailed response and for making clear that there are**
**specific cell type labels with higher classification performance while having lower counts.**
**We would suggest including an r-square value in the regression in Figure 1g. to evaluate**
**the correlation between abundance and classification.**

We thank the reviewer for the suggestion. We’ve added a regression line and an R2 value to the
plot.

**Fig. 1 (g):** F1-score per cell type plotted against the number of unique cells observed per cell type for scTab. The
histogram on the y-axis shows the distribution of F1-scores and the histogram on the x-axis shows the distribution of
unique cells per cell type (log scale).

**4. We thank the authors for addressing this point. It seems like the new Suppl. Table 8 is not**
**referenced in the manuscript.**

We thank the reviewer for pointing this out. We added a reference to the Supp. Table 8 in the main
text:

**Lastly, we show a comparison of the training and inference times in Supp. Table 8.**

**5. We thank the authors for doing a more general comparison of the performance across**
**different organs vs cross-tissue training of the model. We agree that scTab shows a more**
**robust behavior in the cross-tissue context, however, we consider that the lack of label**
**harmonization across tissues could be a factor confounding this result and this should be**
**stated. Also, could the authors please clarify why there is no error bar for specific tissues in**
**the optimized linear bars?**

We thank the reviewers for their comments. For the sake of organization, we split the answer into
two parts:

**We thank the authors for doing a more general comparison of the performance across**
**different organs vs cross-tissue training of the model. We agree that scTab shows a more**
**robust behavior in the cross-tissue context, however, we consider that the lack of label**
**harmonization across tissues could be a factor confounding this result and this should be**
**stated.**

We thank the reviewer for the suggestion. We added the following clarifying remarks to the
manuscript:

Naturally, we must emphasize that inconsistencies between author-provided cell
annotations across tissue systems (and individual researchers) remain a major confounding
factor for performance tests above. These are to be corrected as the community resolves
existing disagreements behind how certain cell types are defined, along with the arrival of
high-quality reference atlases.

**Also, could the authors please clarify why there is no error bar for specific tissues in the**
**optimized linear bars?**

The confidence intervals are missing as they are too small to be displayed. The plot in the main
manuscript is based on the PDF version of the plot which shows the confidence intervals. The plot
we copy pasted in the reviewer's response is based on the PNG version (we just copied the plot
from the jupyter-notebook - as Google Docs doesn't support inserting PDF images). Somehow, the
confidence intervals didn't get displayed properly there. We apologize for missing this.

**6. Thanks for the response. We refer to the answer to point 5 on acknowledging the**
**potential effect of lack of label harmonization on the robustness of the scTab model across**
**tissues, e.g. one of the labels with robust annotation in scTab but not in the optimized linear**
**model is lymphocyte, however, given that this is a very general label, could it be that**
**obtaining a prediction matching more refined labels from a different tissue could be an**
**equally correct or even more helpful result? We understand this requires the**
**time-consuming label harmonization that is not in the scope of this study but consider that**
**it is important to mention it.**

Thank you to the author for these remarks.

At the heart of the issue is that the author-provided cell annotations ingested by CELLxGENE are
inconsistent. We have tried to address the difficulty of this problem in the other responses as well,
as well as the inherent flaws in the training data. The training data does not provide consistent

flawless definitions which could be used as a “true benchmark for biology”; we have tried to
emphasize this fact in the manuscript. That being said, we should note that these CL terms were
selected by the submitting authors themselves, i.e. experts in their respective fields of study. That
alone was motivation for us to use this training data for cell annotation predictions. Once reference
atlases are created which provide a standard for community-consensus behind the definition of cell
types and cell states, such predictions should become less erroneous. We hope that our method
will prove to be of value during the construction of these atlases in the meantime.

**7. We thank the authors for making available the pre-trained models. In connection to the**
**point n.5 from reviewer 4, we would suggest including an application of the pre-trained**
**model(s) to real use-case datasets. To this end, we would suggest providing an example**
**tutorial with a benchmark dataset where also the confidence metrics, such as the above**
**mentioned F1 score, can be visualized. In addition, the application of these pre-trained**
**models to annotate specific disease states, as the dataset employed for the training**
**includes 52 disease states, would be of interest to the community.**

We thank the reviewer for the suggestions. In our response, we would like to proceed by
addressing each point by the reviewer individually:

> **In connection to the point n.5 from reviewer 4, we would suggest including an application**
**of the pre-trained model(s) to real use-case datasets. To this end, we would suggest**
**providing an example tutorial with a benchmark dataset where also the confidence metrics,**
**such as the above mentioned F1 score, can be visualized.**

We understand the motivation behind this request.

However, in practice, it is often impossible to calculate F1-scores for arbitrary datasets as the label
space between the predicted cell types of our scTab model and the label space of the benchmark
dataset rarely overlap. This makes it impossible to directly calculate F1-scores which require a
shared label space between the targets and the predicted labels. That’s also the reason why we
did not add this specific benchmark requested by the reviewer to the tutorial notebook.

> **In addition, the application of these pre-trained models to annotate specific disease**
**states, as the dataset employed for the training includes 52 disease states, would be of**
**interest to the community**

At the moment, we would refrain from using scTab to annotate disease states for two principal
reasons. Firstly, there is only very limited training data from diseased cells available, hence making
it hard to train a reliable classifier. We would like to emphasize here that we merely mentioned in
the paper that the training data contained cells from diseased individuals but never made any
claims that we could predict disease states.

Secondly, disease states of cells remain poorly characterized precisely because the community still
does not have quality reference atlases as a baseline for healthy tissue. Indeed, constructing these
reference atlases for cell types and cell states is the goal of consortia such as the Human Cell
Atlas. Once those reference atlases exist, it is estimated that it will take many years of research to
characterize the transcriptomics of diseased cells across human populations (especially for

intricate diseases such as cancers, which have hundreds of driver mutations each with distinct
transcriptomic signatures.) Given that reality, cell annotations for disease states remain fairly
premature. We would not feel confident promoting such a use case at this stage, and that was
never our intent.

In sum, we thank the reviewer again for their feedback and would like to encourage the reviewer to
open a GitHub Issue or Pull Request if there are any further suggestions to improve the tutorial
notebooks. We would be very happy to incorporate more feedback from the community at large,
and we would be happy to continue the discussion there in the future:
<https://github.com/theislab/scTab>

**8. Thanks to the authors. We refer to our response to point 7.**

**9. We thank the authors for the reasoning of this point. We agree with the response**
**provided.**

**10. We thank the authors for addressing our concern**

*Minor points - Round 2 comments*

**1. We thank the authors for their response. We would suggest including more detail on how**
**cell type labels are distributed across tissues and how they are related in the hierarchy,**
**also, in connection to major point 1.**

We thank the reviewer for their suggestions.

It has been challenging to accommodate such requests given the size of the dimensionality of the
data (e.g. total number of unique tissues) along with the inherent variability within the hierarchies of
the Cell Ontology; often, “deeper” hierarchies do not necessarily correspond to more precise terms
(which is further complicated by the incompleteness of certain cell type lineages). Our aim for the
original subfigure was to give a brief overview of the dataset; more detailed summary statistics
would need far more space or perhaps interactive plots.

To accommodate the first request, we present the original treemap figure with the number of
unique CL terms per tissue superimposed:

**Fig. 1 (a):** Treemap plot showing the composition of the assembled dataset across cell types and tissues. Each of the big
 outer rectangles corresponds to one tissue with the size of the box giving the number of donors for that tissue, the inner
 boxes correspond to the number of donors for each cell type and the color scale highlights the number of unique cell
 types per tissue (the number of unique cell types grouped by Human Cell Atlas bionetworks is shown in Supp. Fig. 1 c).
 The 22.2 million cells from the assembled dataset span 5,052 unique donors, 249 datasets, 52 disease states, 164
 unique cell types, and 56 different tissues (a more detailed overview of the number of donors per cell type and per tissue
 is shown in Supp. Fig. 1; an overview of the number of shared tissues across individual cell types is shown in Supp. Fig.
 2, an interactive visualization of the hierarchical relationship between the cell type labels can be seen in Supp. Fig. 15). A
 full list of the cell type label, tissue, and number of donors and cells per combination of label and tissue is given in Supp.
 Table 5.

Moreover, we added a summary plot showing the number of unique cell types per Human Cell
 Atlas bionetwork to Supp. Fig. 1c:

**Supp. Fig. 1 (c):** Number of unique cell types (CL terms) per Human Cell Atlas bionetwork.

(Note: The tissue labels from CELLxGENE have been aggregated by Human Cell Atlas
BioNetwork in the barplot above. For instance, the BioNetwork 'blood+immune' encompasses the
following tissue labels: 'immune system', 'lymph node', 'blood', 'bone marrow', 'spleen' or the label
'lung' contains the following tissue labels: 'lung', 'respiratory system', 'pleural fluid'. Therefore the
number of unique CL terms might not match the number from the treeplot.)

Regarding the visualization of how cell types are related in the Cell Ontology hierarchy, it is
extremely challenging to visualize the hierarchical relationships between cell type labels in a single
plot. (Please refer to our explanation in Response #2 for why there's no linear relationship between
the cell type relationships/hierarchy and granularity.) We tried to plot the hierarchical relationships
in the graph plot below, as one can see, this results in a messy plot that cannot be displayed in a
non-interactive way:

 **Supp. Figure 15:** Interactive visualization of the hierarchical dependencies of the cell type labels in the scTab training
 dataset.

Scaling such a figure for all CL terms would both not be easily legible, and therefore potentially
 confuse readers more than inform them.

However to accommodate the reviewer comments, we have added an interactive HTML
 visualization of the hierarchy graph to the supplements. Interested readers can study the
 hierarchical dependencies in more detail. We refrained from adding the plot to the main figure due
 to the above mentioned reasons. (There is a niche here for better genomic visualization tools for
 the community, but this is also outside the scope of the current work.)

We've updated the caption of Fig. 1 (a) as follows:

(a) Treemap plot showing the composition of the assembled dataset across cell types and
tissues. Each of the big outer rectangles corresponds to one tissue with the size of the box
giving the number of donors for that tissue, the inner boxes correspond to the number of
donors for each cell type and the color scale highlights the number of unique cell types per
tissue (the number of unique cell types grouped by Human Cell Atlas bionetworks is shown
in Supp. Fig. 1 c). The 22.2 million cells from the assembled dataset span 5,052 unique
donors, 249 datasets, 52 disease states, 164 unique cell types, and 56 different tissues (a
more detailed overview of the number of donors per cell type and per tissue is shown in
Supp. Fig. 1; an overview of the number of shared tissues across individual cell types is
shown in Supp. Fig. 2, an interactive visualization of the hierarchical relationship between
the cell type labels can be seen in Supp. Fig. 15). A full list of the cell type label, tissue, and
number of donors and cells per combination of label and tissue is given in Supp. Table 5.

**2. We thank the authors for this, however, there is still one occurrence in line 786, we ask**
**the authors to please modify.**

We thank the reviewer for spotting this. We've updated the text accordingly.

Reviewer #2 (Remarks to the Author):

I co-reviewed this manuscript with one of the reviewers who provided the listed reports. This is part
of the Nature Communications initiative to facilitate training in peer review and to provide
appropriate recognition for Early Career Researchers who co-review manuscripts.

Author response:

Thank you for your contribution to this round of reviews!

Reviewer #3 (Remarks to the Author):

The authors have addressed most of my comments and concerns. The clarification is very helpful.
Stemming from the discussion of the response letter, I have some clarification questions. Hope it
can be helpful:

Author response:

**1. (From #5) The author mentioned a very interesting point regarding using uncertainty**
**scores to discriminate between correct predictions and the incorrect or newer group. I was**
**wondering if there's any specific guidance for users regarding how to use or interpret these**
**uncertainty scores to do such discrimination.**

Thank you for this question. It's useful to clarify.

We recommend overlaying the uncertainty scores over the plot of the embedding (e.g. UMAP or
tSNE) alongside the predicted cell types for the clusters in question. This will clearly highlight to the
user which model predictions are high confidence vs. low confidence; visualizing the data in this
matter makes it perceptibly easier to spot cells with high uncertainty. As one might expect, cells
with a high uncertainty usually cluster together, making it easy to spot cells/clusters where the
model is uncertain. This can be seen in detail in Supp. Figure 5 a) and b). Based on this
visualization, the user can then select a suitable threshold to select cells for which the model
prediction should be investigated in more detail.

We should be clear that users should investigate the molecular data of each and every prediction,
and decide whether that prediction is accurate based on biological priors (e.g. the expertise of the
biologist doing the annotations). The uncertainty scores are vital simply to help guide the biologist
as to which predictions require the standard workflow of manual annotation (which includes
investigating the differentially expressed genes of a given cluster, deciding the clustering
resolution, and so on).

We've added the following clarifying remarks to the manuscript:

In practice, biologists and computational biologists can overlay the uncertainty estimates
from scTab alongside the predicted cell type labels and their defined clustering on a UMAP
or tSNE visualization of their data to hint at which predictions are associated with higher
uncertainty and, hence, should be investigated in more detail (see Supp. Fig. 5a).

**2. (From #6) The author mentions in the method section a filtering step for selecting a more**
**focused group of augmentation vectors to ensure they are insensitive to the cell types. I**
**was wondering if the authors could clarify with more details about how such a procedure is**
**performed so that the users can perform in practice. For example, how K is determined by**
**such a K-means clustering?**

We thank the reviewer for this question. Indeed, this is an interesting point that we could expand
upon. We recommend treating the resulting parameters in the same way as one would treat the
hyperparameters of a neural network: that is to say, one calculates the augmentation vectors
based on the training split, and then evaluates how well the selected hyperparameters work based
on the performance on the validation set. As a final step, you report the performance on the
holdout test set to make sure that there is no overfitting with respect to the chosen
hyperparameters.

Moreover, we would like to note that the performance seems to be quite robust with respect to the
hyperparameters of our filtering step. We've mostly left the parameters at our first initial guess
untouched as those parameters already achieved satisfactory performance. Further tuning of the
associated hyperparameters would be an interesting direction for future work.

We've added the following clarifying remarks to the manuscript:

The associated parameters of our data augmentation strategy should be tuned in a similar
fashion as one would tune the hyperparameter of a neural network. This means calculating
the augmentation vectors based on the training split, then selecting the best parameter set
based on the validation split and finally, reporting the performance on the holdout test set.

Reviewer #4 (Remarks to the Author):

The response to my review reflects a thoughtful consideration of feedback, resulting in substantial
enhancements to the manuscript. The authors' revisions significantly bolstered clarity and
effectively addressed key concerns. The paper introduces scTab, a novel approach for cell type
prediction in single-cell transcriptomics. I recommend acceptance for publication based on the
manuscript's improved quality and its potential to propel cell type annotation in the field of
single-cell transcriptomics.

Author response:

Thank you.

Reviewer #1 (Remarks to the Author):

We thank the authors for going the extra mile to address all reviewer comments and congratulate them for an excellent piece of work. We are happy to recommend acceptance of the manuscript.